# DIFFUSION-BASED PLANNING FOR AUTONOMOUS DRIVING WITH FLEXIBLE GUIDANCE

**Yinan Zheng**[1][*], **Ruiming Liang**[2][*][‡], **Kexin Zheng**[3][*][‡], **Jinliang Zheng**[1], **Liyuan Mao**[4][‡],
**Jianxiong Li**[1], **Weihao Gu**[5], **Rui Ai**[5], **Shengbo Eben Li**[1], **Xianyuan Zhan**[1,6][†] , **Jingjing Liu**[1][†]
[1] Tsinghua University     [2] Institute of Automation, Chinese Academy of Sciences
[3] The Chinese University of Hong Kong     [4] Shanghai Jiao Tong University
[5] HAOMO.AI    [6] Shanghai Artificial Intelligence Laboratory
zhengyn23@mails.tsinghua.edu.cn, zhanxianyuan@air.tsinghua.edu.cn

## ABSTRACT

Achieving human-like driving behaviors in complex open-world environments is a critical challenge in autonomous driving. Contemporary learning-based planning approaches such as imitation learning methods often struggle to balance competing objectives and lack of safety assurance, due to limited adaptability and inadequacy in learning complex multi-modal behaviors commonly exhibited in human planning, not to mention their strong reliance on the fallback strategy with predefined rules. We propose a novel transformer-based *Diffusion Planner* for closed-loop planning, which can effectively model multi-modal driving behavior and ensure trajectory quality without any rule-based refinement. Our model supports joint modeling of both prediction and planning tasks under the same architecture, enabling cooperative behaviors between vehicles. Moreover, by learning the gradient of the trajectory score function and employing a flexible classifier guidance mechanism, *Diffusion Planner* effectively achieves safe and adaptable planning behaviors. Evaluations on the large-scale real-world autonomous planning benchmark nuPlan and our newly collected 200-hour delivery-vehicle driving dataset demonstrate that *Diffusion Planner* achieves state-of-the-art closed-loop performance with robust transferability in diverse driving styles. Project website: https://zhengyinan-air.github.io/Diffusion-Planner/.

## 1 INTRODUCTION

Autonomous driving as a cornerstone technology, is poised to usher transportation into a safer and more efficient era of mobility (Tampuu et al., 2020). The key challenge is achieving human-like driving behaviors in complex open-world environment, while ensuring safety, efficiency, and comfort (Muhammad et al., 2020). Rule-based planning methods have demonstrated initial success in industrial applications (Fan et al., 2018), by defining driving behaviors and establishing boundaries derived from human knowledge. However, their reliance on predefined rules limits adaptability to new traffic situations (Hawke et al., 2020), and modifying rules demands extensive engineering effort. In contrast, learning-based planning methods acquire driving skills by cloning human driving behaviors from collected datasets (Caesar et al., 2021), a process made simpler through straightforward imitation learning losses. Additionally, the capabilities of these models can potentially be enhanced by scaling up training resources (Chen et al., 2023).

Though promising, current learning-based planning methods still face several limitations. Firstly, human drivers often exhibit multi-modal behaviors in planning scenarios(Nayakanti et al., 2023). Existing methods that rely on behavior cloning lack a guarantee of fitting such complex data distributions, even when utilizing large transformer-based model architecture or sampling multiple trajectories (Cheng et al., 2023). Secondly, when encountering out-of-distribution (OOD) scenarios, directly using model output may result in low-quality planning outcomes, forcing many methods to

---

[*]Equal contribution.
[†]Corresponding authors.
[‡]Work done during internships at Institute for AI Industry Research (AIR), Tsinghua University.

fall back on rule-based approaches for trajectory refinement optimization or filtering (Vitelli et al., 2022; Huang et al., 2023), inevitably facing the same inherent limitations associated with rule-based methods. Thirdly, imitation learning alone is inadequate to capture the vast diversity of driving behaviors required for autonomous driving. For example, penalizing unsafe planning via auxiliary loss, as employed in existing methods (Bansal et al., 2018; Cheng et al., 2024), often results in multi-objective conflicts and poor safety performance due to the lack of learning signals that can teach the agent to recover from mistakes (Zheng et al., 2024; Chen et al., 2021). Additionally, well-trained models may be difficult to adapt behaviors to meet specific needs.

In this study, we discover that diffusion model (Ho et al., 2020) possesses huge potential to address the aforementioned issues. Its ability to model complex data distributions (Chi et al., 2023) allows for effective capturing of multi-modal human driving behavior. Additionally, the high-quality generation capability of the diffusion model also provides opportunities for improving the output trajectory quality through appropriate structural design, removing the reliance on rule-based refinement. The best part of diffusion lies in its flexible guidance mechanism (Dhariwal & Nichol, 2021), which allows adaptation to various planning behavioral needs without additional training. Inspired by these observations, we introduce a novel learning-based approach, *Diffusion Planner*, which pioneers the use of diffusion models (Ho et al., 2020) for enhancing closed-loop planning performance without any rule-based refinement. *Diffusion Planner* is realized by learning the gradient of vehicles' trajectory score function (Song & Ermon, 2019) to model the multi-modal data distribution, and further enables personalized planning behavior adaptation through a classifier guidance mechanism. Specifically, we propose a new network architecture built upon the diffusion transformer (Peebles & Xie, 2023). The diffusion loss is employed to jointly train both prediction and planning tasks within the same architecture, enabling cooperative behaviors between vehicles without the need for additional loss functions. Moreover, the versatility of classifier guidance is further demonstrated by its ability to modify the planning behavior of the trained model, such as enhancing safety and comfort, or controlling the vehicle's speed. The differentiable classifier score can be computed in parallel and is flexible for combination, without requiring additional training. Evaluation results on the large-scale real-world autonomous planning benchmark nuPlan (Caesar et al., 2021) demonstrate that *Diffusion Planner* achieves state-of-the-art closed-loop performance among learning-based baselines, comparable to or even surpassing rule-based methods, directly using the model's output without any additional post-processing. By appending a existing post-processing module to the model, we further achieved state-of-the-art performance among all baselines. Additionally, we collected 200 hours of long-term delivery-vehicle driving data in various city-driving scenarios that further validate the transferability and robustness of the model in diverse driving styles.

In summary, our contributions are:

- To the best of our knowledge, we are the first to fully harness the power of diffusion models with a specifically designed architecture for high-performance motion planning, without overly reliant on rule-based refinement.

- We achieve state-of-the-art performance on the real-world nuPlan dataset, generating more robust and smoother trajectories compared to the baselines.

- We demonstrate that our model can achieve personalized driving behavior at runtime by utilizing a flexible guidance mechanism, which is a desirable feature for real-world applications.

- We have collected and evaluated a new 200-hour delivery-vehicle dataset, which is compatible with the nuPlan framework, and we will open-source it.

## 2 RELATED WORK

**Rule-based Planner**. Rule-based methods rely on predefined rules to dictate the driving behavior of autonomous vehicles, offering a highly controllable and interpretable decision-making process (Treiber et al., 2000a; Fan et al., 2018; Dauner et al., 2023a). While they have been widely validated in real-world scenarios (Leonard et al., 2008; Urmson et al., 2008), these frameworks are limited in their ability to handle novel complex situations that fall beyond the predefined rules.

**Learning-based Planner**. Learning-based planning focuses on leveraging methods such as behavior cloning in imitation learning to directly model human driving behaviors, which has emerged as a popular solution in autonomous driving, particularly in recent end-to-end training pipelines (Hu

et al., 2023; Tampuu et al., 2020; Chen et al., 2023). Behavior cloning method was initially implemented using CNN (Bojarski et al., 2016; Kendall et al., 2019; Hawke et al., 2020) or RNN (Bansal et al., 2018) networks and has since been extended to Transformer due to its strong performance and efficiency in fitting complex data distributions (Scheel et al., 2021; Chitta et al., 2022). However, these methods lack theoretical guarantees for modeling multi-modal driving behavior, which can lead to serious error accumulation in closed-loop planning. As a result, most existing approaches still heavily rely on rules to refine (Vitelli et al., 2022; Huang et al., 2023) or select (Cheng et al., 2024) the generated trajectories, which in some sense, has failed their initial purpose of using learning to replace pre-defined rules. While learning-based methods could offer more human-like driving behavior, their uncontrollable outputs lack safety guarantees and are hard to adjust based on user needs. Existing methods add extra training losses (Bansal et al., 2018; Cheng et al., 2024), but struggle to strike a balance among competing learning objectives. Additionally, these methods also lack flexibility, making post-training behavior adjustments difficult. In practice, it is desirable for a trained planning model to achieve flexible alignment to various safety and personalized driving preferences during inference, which is still lacking in the current literature. In this work, we develop a novel diffusion planner to tackle the above limitations, which enables the generation of high quality planning trajectories without the need for rule-based refinement, and flexible post-training adaptation to various driving styles through the diffusion guidance mechanism.

**Diffusion-based Methods Used in Related Domain**. Diffusion models have been recently explored in decision-making fields (Janner et al., 2022; Chi et al., 2023; Liu et al., 2025), however, their use in autonomous planning has not yet been fully explored. Some existing works employ diffusion models for motion prediction (Jiang et al., 2023) and traffic simulation (Zhong et al., 2023b;a), but their focus is on open-loop performance or diversity in simulation rather than quality or drivability, as the outputs are not directly used for control. There are also studies targeting planning (Hu et al., 2024; Yang et al., 2024; Sun et al., 2023), but these approaches only apply diffusion loss to existing frameworks or stack parameters without specific design considerations, making them heavily reliant on post-processing for reasonable performance. In this paper, we demonstrate that with appropriate structural design, the potential of diffusion models can be fully harnessed to enhance closed-loop planning performance in autonomous driving.

## 3 PRELIMINARIES

### 3.1 AUTONOMOUS DRIVING AND CLOSED-LOOP PLANNING

The primary objective of autonomous driving is to allow vehicles to navigate complex environments with minimal human intervention, where a critical challenge is closed-loop planning (Caesar et al., 2021). Unlike open-loop planning (Caesar et al., 2019) or motion prediction (Ngiam et al., 2021; Zhou et al., 2023), which only involves decision making that adapts to static conditions, closed-loop planning requires a seamless integration of real-time perception, prediction, and control. Vehicles must continuously assess their surroundings, predict the behavior of other neighboring vehicles, and implement precise maneuvers. The dynamic nature of real-world driving scenarios, combined with uncertainty in sensor data and environmental factors, makes closed-loop planning a formidable task.

### 3.2 DIFFUSION MODEL AND GUIDANCE SCHEMES

**Diffusion Model**. Diffusion Probabilistic Models (Sohl-Dickstein et al., 2015; Ho et al., 2020) are a class of generative models that generate outputs by reversing a Markov chain process known as the forward diffusion process. The transition distribution of the forward process satisfies:

$$q_{t0}(\boldsymbol{x}^{(t)}|\boldsymbol{x}^{(0)}) = \mathcal{N}(\boldsymbol{x}^{(t)} \mid \alpha_t \boldsymbol{x}^{(0)}, \sigma_t^2 \mathbf{I}), t \in [0,1], \tag{1}$$

which gradually adds Gaussian noise to generate a series of noised data from $\boldsymbol{x}^{(0)}$ to $\boldsymbol{x}^{(t)}$ with $t \in [0,1]$. $\sigma_t > 0$ is a variance term that controls the introduced noise and $\alpha_t > 0$ is typically defined as $\alpha_t = \sqrt{1 - \sigma_t^2}$, ensuring $\boldsymbol{x}^{(t)} \to \mathcal{N}(0, \mathbf{I})$, as $t \to 1$. The reversed denoising process of Eq. (1) can be equivalently expressed as a diffusion ODE (Song et al., 2021):

$$\text{(Diffusion ODE)} \quad \mathrm{d}\boldsymbol{x}^{(t)} = \left[ f(t)\boldsymbol{x}^{(t)} - \frac{1}{2}g^2(t)\nabla_{\boldsymbol{x}^{(t)}} \log q_t(\boldsymbol{x}^{(t)}) \right] \mathrm{d}t, \tag{2}$$

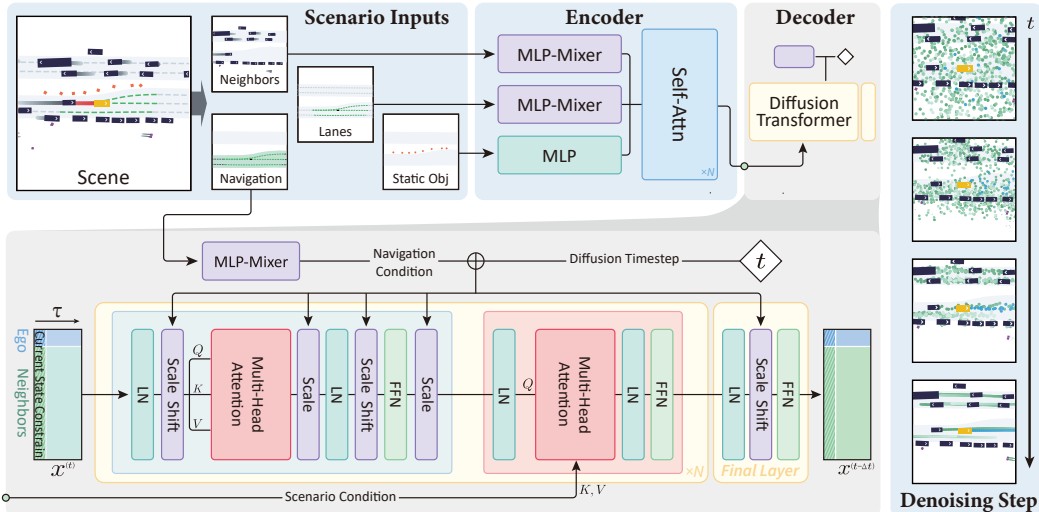

Figure 1: Model architecture of *Diffusion Planner*.

where $f(t) = \frac{\mathrm{d}\log\alpha_t}{\mathrm{d}t}, g^2(t) = \frac{\mathrm{d}\sigma_t^2}{\mathrm{d}t} - 2\frac{\mathrm{d}\log\alpha_t}{\mathrm{d}t}\sigma_t^2$ are determined by the fixed noise schedules $\alpha_t, \sigma_t$, and $q_t$ is the marginal distribution of $\boldsymbol{x}^{(t)}$. Diffusion model utilizes a neural network $\boldsymbol{s}_\theta(\boldsymbol{x}^{(t)}, t)$ to fit the probability score $\nabla_{\boldsymbol{x}^{(t)}} \log q_t(\boldsymbol{x}^{(t)})$. By learning the score function, diffusion models enjoy the strong expressiveness of modeling arbitrary complex distributions (Chi et al., 2023), making it highly versatile and adaptable for challenging tasks such as autonomous driving.

**Classifier Guidance**. Classifier guidance (Dhariwal & Nichol, 2021) is a technique used to generate preferred data by guiding the sampling process with a classifier $\mathcal{E}_\phi(\boldsymbol{x}^{(t)}, t)$. The gradient of the classifier score is used to modify the original diffusion score:

$$\tilde{\boldsymbol{s}}_\theta(\boldsymbol{x}^{(t)}, t) = \boldsymbol{s}_\theta(\boldsymbol{x}^{(t)}, t) - \nabla_{\boldsymbol{x}^{(t)}} \mathcal{E}_\phi(\boldsymbol{x}^{(t)}, t) \tag{3}$$

In autonomous driving, this approach offers greater flexibility compared to rule-based refinement because it directly improves the model's inherent ability, rather than overly relying on sub-optimal post-processing that requires significant human effort and targeted data collection.

## 4 METHODOLOGY

In this section, we redefine the planning task as a future trajectory generation task, which jointly generates the ego vehicle's planning and the prediction of neighboring vehicles. We then introduce the *Diffusion Planner*, a novel approach that leverages the expressive and flexible diffusion model for enhanced autonomous planning. Lastly, we demonstrate how the guidance mechanism in diffusion models can be utilized to align planning behavior with safe or human-preferred driving styles.

### 4.1 TASK REDEFINITION

Autonomous driving requires considering the close interaction between the ego and neighboring vehicles, resulting in a cooperative relationship between planning and motion prediction tasks (Ngiam et al., 2021). Supervising the future trajectories of neighboring vehicles has been shown to be helpful to enhance the ability of closed-loop planning models to handle complex interaction scenarios (Hu et al., 2023). For real-world deployment, motion prediction can also enhance safety by providing more controllable measures, facilitating the implementation of the system (Fan et al., 2018). Consequently, the trajectories of neighboring vehicles have become crucial privileged information for model training. However, the common approaches that use a dedicated sub module (Huang et al., 2023) or additional loss design (Cheng et al., 2023; Huang et al., 2023) to capture privileged information limit their modeling power during training and also lead to a more complex framework.

In this work, we address this issue by collectively considering the status of key participants in the driving scenario and jointly modeling the motion prediction and closed-loop planning tasks as a

*future trajectory generation* task. Specifically, given conditions $\boldsymbol{C}$, which include current vehicle states, historical data, lane information, and navigation information, our goal is to generate future trajectories for all key participants simultaneously, enabling the modeling of cooperative behaviors among them. However, this joint modeling of complex distributions is challenging to solve with a simple behavior cloning approach. Benefiting from the strong expressive power of diffusion models, we adopt a diffusion model for this task and formulate the target as:

$$\boldsymbol{x}^{(0)} = \begin{bmatrix} x_{\text{ego}}^{(0)} \\ x_{\text{neighbor}_1}^{(0)} \\ \vdots \\ x_{\text{neighbor}_M}^{(0)} \end{bmatrix} = \begin{bmatrix} x_{\text{ego}}^1 & x_{\text{ego}}^2 & \cdots & x_{\text{ego}}^\tau \\ x_{\text{neighbor}_1}^1 & x_{\text{neighbor}_1}^2 & \cdots & x_{\text{neighbor}_1}^\tau \\ \vdots & \vdots & \ddots & \vdots \\ x_{\text{neighbor}_M}^1 & x_{\text{neighbor}_M}^2 & \cdots & x_{\text{neighbor}_M}^\tau \end{bmatrix}, \tag{4}$$

where we use superscripts with parentheses to represent the timeline of diffusion denoising, and regular superscripts to indicate the timeline of the future trajectory, which contains $\tau$ time steps. For each state $x$, we only consider the coordinates and the sine and cosine of the heading angle, which are sufficient for the downstream LQR controller. We select the nearest $M$ neighboring vehicles to predict their possible future trajectories. By parameterizing our *Diffusion Planner* with $\theta$, the training target can be expressed as:

$$\mathcal{L}_\theta = \mathbb{E}_{\boldsymbol{x}^{(0)}, t \sim \mathbb{U}(0,1), \boldsymbol{x}^{(t)} \sim q_{t0}(\boldsymbol{x}^{(t)}|\boldsymbol{x}^{(0)})} \left[ ||\mu_\theta(\boldsymbol{x}^{(t)}, t, \boldsymbol{C}) - \boldsymbol{x}^{(0)}||^2 \right], \tag{5}$$

where the goal is to recover the data distribution from noisy data (Ramesh et al., 2022). We can get the score function as $\boldsymbol{s}_\theta = (\alpha_t \mu_\theta - \boldsymbol{x}^{(t)})/\sigma_t^2$ and apply it during the denoising process. The joint prediction of multiple vehicles is similar to motion prediction (Jiang et al., 2023) and traffic simulation (Zhong et al., 2023b;a) tasks, but we focus more on the ego vehicle's closed-loop planning performance and real-time deployment. We will introduce the specific designs as follows.

## 4.2 DIFFUSION PLANNER

*Diffusion Planner* is a model based on the DiT architecture (Peebles & Xie, 2023), with a core design focusing on the fusion mechanism between noised future vehicle trajectories $\boldsymbol{x}$ and conditional information $\boldsymbol{C}$. Figure 1 provides an overview of the complete architecture. A detailed description of these interaction and fusion modules is provided as follows.

**Vehicle Information Integration**. In the first step, the future vehicle trajectory $\boldsymbol{x}$ is concatenated with the current state of each vehicle, represented as $x^0 = [x_{\text{ego}}^0, x_{\text{neighbor1}}^0, \ldots, x_{\text{neighbor}_M}^0]^T$. This concatenation acts as a constraint to guide the model, simplifying the planning task by providing a clear starting point. Notably, velocity and acceleration information for the ego vehicle is excluded, which has been shown to enhance closed-loop performance, as highlighted in previous works (Cheng et al., 2023; Li et al., 2024). Integration of the information from different vehicles during model execution is achieved through multi-head self-attention mechanisms.

**Historical Status and Lane Information Fusion**. The historical status of neighboring vehicles and lane information is represented using vectors (Gao et al., 2020). Specifically, each neighboring vehicle is represented as $\boldsymbol{S}_{\text{neighbor}} \in \mathbb{R}^{L \times D_{\text{neighbor}}}$, and lanes as $\boldsymbol{S}_{\text{lane}} \in \mathbb{R}^{P \times D_{\text{lane}}}$, where $L$ refers to the number of past timestamps, and $P$ indicates the number of points per polyline. $D_{\text{neighbor}}$ contains data such as vehicle coordinates, heading, velocity, size, and category, while $D_{\text{lane}}$ provides lane details such as coordinates, traffic light status, and speed limits. Since these vectors are information-sparse, directly fusing them would make training challenging. To address this, we use MLP-Mixer network (Tolstikhin et al., 2021) to extract information-dense representations. Compared to existing work (Huang et al., 2023; Cheng et al., 2023) that uses complex structural designs, we offer a more unified and simplified solution. This is achieved by iteratively passing the vectors through the MLP mixing layers, which operate on both the vector and feature dimensions. The forward process of each mixing layer can be formulated as follows:

$$\boldsymbol{S} = \boldsymbol{S} + \text{MLP}(\boldsymbol{S}^T)^T, \boldsymbol{S} = \boldsymbol{S} + \text{MLP}(\boldsymbol{S}) \tag{6}$$

We use two separate MLP-Mixer networks for neighboring vehicles and lanes. Here, $\boldsymbol{S}$ represents the features for each neighboring vehicle or lane. After passing through multiple mixing layers, we apply pooling on the final output along the vector dimension. We also consider the static objects

information $\boldsymbol{S}_{\text{static}} \in \mathbb{R}^{D_{\text{static}}}$, where $D_{\text{static}}$ includes data such as coordinates, heading, size, and category. For this, we use an MLP to extract the representation. Finally, we concatenate all representations and feed them into a vanilla transformer encoder for further aggregation, resulting in the encoder representation $\boldsymbol{Q}_f$. The fusion of $\boldsymbol{Q}_f$ with $\boldsymbol{x}$ proceeds as follows:

$$\boldsymbol{x} = \boldsymbol{x} + \text{MHCA}(\boldsymbol{x}, \boldsymbol{Q}_f), \boldsymbol{x} = \boldsymbol{x} + \text{FFN}(\boldsymbol{x}), \tag{7}$$

where MHCA donate multi-head cross-attention.

**Navigation Information Fusion**. Navigation information is crucial for autonomous driving planning, as it provides essential guidance on the intended route, enabling the vehicle to make informed decisions. In the nuPlan benchmark (Caesar et al., 2021), navigation information is represented as a set of lanes along a route, $\boldsymbol{S}_{\text{route}} \in \mathbb{R}^{(K \times P) \times D_{\text{route}}}$, where $K$ denotes the number of route lanes, and $D_{\text{route}}$ contains only coordinate information. We first employ an MLP-Mixer network, as described in equation 6, to extract the essential guidance representations $\boldsymbol{Q}_n$. $\boldsymbol{Q}_n$ is then added to the diffusion timestep condition $\boldsymbol{Q}_t$ and applied through an adaptive layer norm block (Peebles & Xie, 2023) to guide trajectory generation across all tokens.

## 4.3 PLANNING BEHAVIOR ALIGNMENT VIA CLASSIFIER GUIDANCE

Enforcing versatile and controllable driving behavior is crucial for real-world autonomous driving. For example, vehicles must ensure safety and comfort while adjusting speeds to align with user preferences. Thanks to its close relationship to Energy-Based Models (Lu et al., 2023), diffusion model can conveniently inject such preferences via classifier guidance. It can steer the model outputs via gradient surgery during inference, offering significant potential for customized adaptation.

Specifically, given the original driving behavior $q_0(\boldsymbol{x}^{(0)})$, we aim to encode additional guidance to reinforce some preferred behavior upon the existing behavior $q_0$. This operation can be formulated as generating a target behavior: $p_0(\boldsymbol{x}^{(0)}) \propto q_0(\boldsymbol{x}^{(0)}) e^{-\mathcal{E}(\boldsymbol{x}^{(0)})}$, where $\mathcal{E}(\boldsymbol{x}^{(0)})$ can be some form of energy function that encodes safety or preferred behavior. As mentioned in Section 3.2, the gradient of the intermediate energy (Lu et al., 2023) is employed to adjust

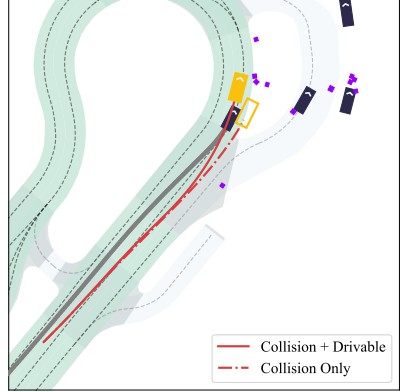

Figure 2: Starting from the same position, the trajectories driven under different guidance settings.

the original probability score, promoting the generation of trajectories within the target distribution. This process often necessitates an additional trained classifier to provide an accurate approximation. However, diffusion posterior sampling (Chung et al., 2022; Xu et al., 2025) offers a training free method that only uses the trained diffusion model $\mu_\theta$ in Eq. (5) to approximate the guidance energy, bypassing the classifier training, which incurs additional computational overhead:

$$\begin{aligned} \nabla_{\boldsymbol{x}^{(t)}} \log p_t(\boldsymbol{x}^{(t)}) &\approx \nabla_{\boldsymbol{x}^{(t)}} \log q_t(\boldsymbol{x}^{(t)}) - \nabla_{\boldsymbol{x}^{(t)}} \mathcal{E} \left( \mathbb{E}_{q_{0t}(\boldsymbol{x}^{(0)} | \boldsymbol{x}^{(t)})} [\boldsymbol{x}^{(0)}] \right) \\ &= \nabla_{\boldsymbol{x}^{(t)}} \log q_t(\boldsymbol{x}^{(t)}) - \nabla_{\boldsymbol{x}^{(t)}} \mathcal{E} \left( \mu_\theta(\boldsymbol{x}^{(t)}, t, \boldsymbol{C}) \right). \end{aligned} \tag{8}$$

One restriction of this method is that Eq. (8) needs to use a pre-defined differentiable energy function $\mathcal{E}(\cdot)$ to calculate the guidance energy. Fortunately, in autonomous driving scenarios, many trajectory evaluation protocols can be defined using differentiable functions. Next, we briefly describe some applicable energy functions that can be used to customize the planning behavior of the model, more details are shown in Appendix C.3.

- **Target speed maintenance**: The speed difference is used as the energy, calculated by comparing the planned average speed with the set target speed.
- **Comfort**: The energy function is calculated by measuring the amount by which the vehicle's state exceeds the predefined limits.
- **Collision avoidance**: The signed distance between the ego vehicle and neighboring vehicles is computed at each timestamp.

- **Staying within drivable area**: The distance the ego vehicle deviates outside the lane at each time step is calculated.

Additionally, this training-free approach supports flexible combinations during inference time, providing a solution for controllable trajectory generation in complex scenarios. For example, as shown in Figure 2, under collision guidance alone, the ego vehicle veers off the road to avoid a rear-approaching vehicle. However, when drivable guidance is added, the vehicle stays on the road while maintaining safety. For more case studies, please refer to Section 5.1 and the Appendix B.2.

## 4.4 PRACTICAL IMPLEMENTATION FOR CLOSED-LOOP PLANNING

Data augmentation can help alleviate the out-of-distribution issue and is widely used in planning. Before training, we add random perturbations to the current state (Cheng et al., 2023). Then, interpolation is applied to create a physically feasible transition, enabling the model to resist perturbations and regress to the ground-truth trajectory (Bansal et al., 2018). After that, we transform the data from the global coordinate system into an ego-centric formulation through coordinate transformation. Considering the significant difference between the longitudinal and lateral distances traveled by the vehicle, z-score normalization is used to ensure the mean of the data distribution is close to zero, thereby further stabilizing the training process. During inference, DPM-Solver (Lu et al., 2022) is employed to achieve faster sampling, while low-temperature sampling (Ajay et al., 2022) enhances determinism in the planning process. We can complete trajectory planning for the next 8 seconds at 10 Hz, along with predictions for neighboring vehicles, with an inference frequency of approximately 20 Hz. Please see Appendix C for implementation details.

## 5 EXPERIMENTS

**Evaluation Setups**. We conduct extensive evaluations on the large-scale real-world autonomous planning benchmark, nuPlan (Caesar et al., 2021), to compare *Diffusion Planner* with other state-of-the-art planning methods. The Val14 (Dauner et al., 2023b), Test14, and Test14-hard benchmarks (Cheng et al., 2023) are utilized, with all experimental results tested in both closed-loop non-reactive and reactive modes. The final score is calculated as the average across all scenarios, ranging from 0 to 100, where a higher score indicates better algorithm performance. To further validate the algorithm's performance across diverse driving scenarios and with vehicles exhibiting different driving behaviors, we collected 200 hours of real-world data using a delivery vehicle from Haomo.AI. Unlike nuPlan, the delivery vehicle demonstrates more conservative planning behavior and operates in bike lanes, which involve dense human-vehicle interactions and unique traffic regulations. The collected data were integrated into the nuPlan framework, and the same evaluation metrics were applied in closed-loop simulations, as detailed in Appendix D.

**Baselines**. The baselines are categorized into three groups (Dauner et al., 2023b): *Rule-based*, *Learning-based*, and *Hybrid*, which incorporate additional refinement to the outputs of the learning-based model. To enable a more comprehensive comparison, we utilize an existing refinement module (Sun et al., 2024), which applies offsets to the model outputs and scores all trajectories (Dauner et al., 2023b). Without any parameter tuning, we integrate this module as post-processing for the *Diffusion Planner* (*Diffusion Planner w/ refine.*). We compare the *Diffusion Planner* against the following baselines, with more implementation details provided in Appendix C.4.

- *IDM* (Treiber et al., 2000b): A classic rule-based method implemented by nuPlan.

- *PDM* (Dauner et al., 2023b): The first-place winner of the nuPlan challenge offers a rule-based version that follows the centerline (*PDM-Closed*), a learning-based version conditioned on the reference line (*PDM-Open*), and a hybrid approach that combines both (*PDM-Hybrid*).

- *UrbanDriver* (Scheel et al., 2021): A learning-based method using policy gradient optimization and implemented by nuPlan.

- *GameFormer* (Huang et al., 2023): Modeling ego and neighboring vehicle interactions using game theory (*GameFormer w/o refine.*), followed by rule-based refinement.

- *PlanTF* (Cheng et al., 2023): A state-of-the-art learning-based method built on a transformer architecture, exploring various designs suitable for closed-loop planning.

Table 1: Closed-loop planning results on nuPlan dataset. ▨: The highest scores of baselines in various types. *: Using pre-searched reference lines as model input provides prior knowledge, reducing the difficulty of planning compared to standard learning-based methods. NR: non-reactive mode. R: reactive mode.

| Type | Planner | Val14 | | Test14-hard | | Test14 | |
|---|---|---|---|---|---|---|---|
| | | NR | R | NR | R | NR | R |
| Expert | Log-replay | 93.53 | 80.32 | 85.96 | 68.80 | 94.03 | 75.86 |
| Rule-based & Hybrid | IDM | 75.60 | 77.33 | 56.15 | 62.26 | 70.39 | 74.42 |
| | PDM-Closed | 92.84 | 92.12 | 65.08 | 75.19 | 90.05 | 91.63 |
| | PDM-Hybrid | 92.77 | 92.11 | 65.99 | 76.07 | 90.10 | 91.28 |
| | GameFormer | 79.94 | 79.78 | 68.70 | 67.05 | 83.88 | 82.05 |
| | PLUTO | 92.88 | 76.88 | 80.08 | 76.88 | 92.23 | 90.29 |
| | Diffusion Planner w/ refine. (Ours) | 94.26 | 92.90 | 78.87 | 82.00 | 94.80 | 91.75 |
| Learning-based | PDM-Open* | 53.53 | 54.24 | 33.51 | 35.83 | 52.81 | 57.23 |
| | UrbanDriver | 68.57 | 64.11 | 50.40 | 49.95 | 51.83 | 67.15 |
| | GameFormer w/o refine. | 13.32 | 8.69 | 7.08 | 6.69 | 11.36 | 9.31 |
| | PlanTF | 84.27 | 76.95 | 69.70 | 61.61 | 85.62 | 79.58 |
| | PLUTO w/o refine.* | 88.89 | 78.11 | 70.03 | 59.74 | 89.90 | 78.62 |
| | Diffusion Planner (Ours) | 89.87 | 82.80 | 75.99 | 69.22 | 89.19 | 82.93 |

Table 2: Closed-loop planning results on delivery-vehicle driving dataset.

| Type | Planner | Score | Collisions | TTC | Drivable | Comfort | Progress |
|---|---|---|---|---|---|---|---|
| Rule-based | IDM | 75.38 | 86.00 | 79.43 | 99.43 | 89.14 | 95.43 |
| | PDM-Closed | 80.95 | 86.51 | 80.00 | 100.0 | 97.21 | 97.47 |
| Hybrid | PDM-Hybrid | 80.72 | 86.50 | 77.00 | 100.0 | 92.50 | 99.00 |
| | GameFormer | 51.35 | 82.50 | 72.50 | 65.00 | 98.00 | 90.00 |
| | PLUTO | 83.49 | 88.95 | 85.64 | 99.45 | 94.47 | 97.79 |
| Learning-based | PDM-Open* | 64.84 | 75.75 | 70.50 | 93.50 | 98.50 | 95.00 |
| | GameFormer w/o refine. | 22.41 | 62.00 | 57.50 | 33.00 | 98.50 | 77.00 |
| | PlanTF | 90.89 | 95.00 | 90.50 | 99.50 | 96.00 | 99.50 |
| | PLUTO w/o refine. | 87.77 | 92.69 | 87.64 | 99.44 | 97.19 | 98.88 |
| | Diffusion Planner (ours) | 92.08 | 96.00 | 91.00 | 100.0 | 94.00 | 100.0 |

| Diffusion Planner (ours) | GameFormer w/o refine. | PlanTF | PLUTO w/o refine. |
|---|---|---|---|

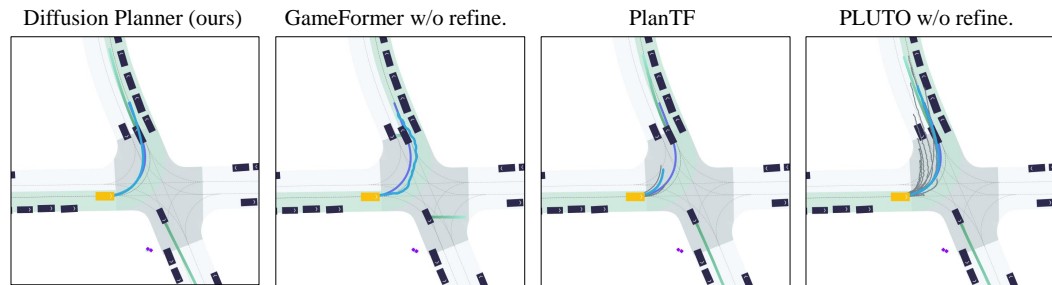

Figure 3: Future trajectory generation visualization. A frame from a challenging narrow road turning scenario in the closed-loop test, including the future planning of the ego vehicle (*PlanTF* and *PLUTO w/o refine.* showing multiple candidate trajectories), predictions for neighboring vehicles, and the ground truth ego trajectory.

- *PLUTO* (Cheng et al., 2024): Building on *PDM-Open*, a complex model with contrastive learning enhances environmental understanding (*PLUTO w/o refine.*), followed by post-processing.

**Main Results**. Evaluation results on the nuPlan benchmark are presented in Table 1. The *Diffusion Planner* achieves state-of-the-art performance across more benchmarks compared to all learning-based baselines. With the addition of post-processing, *Diffusion Planner w/ refine.* outperforms hybrid and rule-based baselines, achieving scores that even surpass human performance. This is due to our model's ability to output high-quality trajectories, which are further enhanced by post-processing. Notably, compared to the transformer-based *PlanTF* and *PLUTO*, *Diffusion Planner* leverages the power of diffusion to achieve better performance. *GameFormer*, which models the

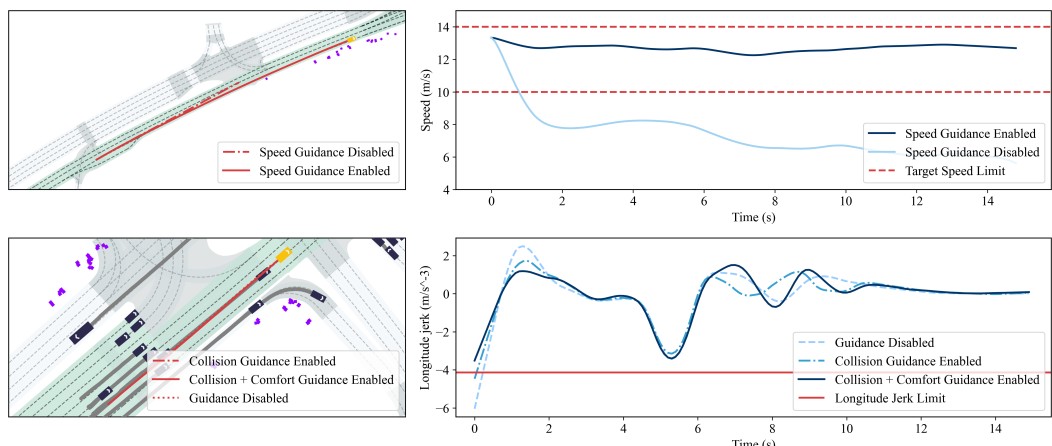

Figure 4: Target speed and comfort guidance: For target speed guidance, the speed changes before and after guidance are visualized. For comfort guidance, the longitudinal jerk changes are compared before and after applying comfort guidance on top of collision avoidance guidance.

interactions between the ego vehicle and neighboring vehicles using game theory, exhibits limited model capabilities, making it overly reliant on rule-based refinements. We further present the planning results on delivery-vehicle driving dataset as shown in Table 2. *PDM*, *GameFormer*, and *PLUTO* include certain designs specifically tailored to the nuPlan benchmark, which limits their ability to transfer to delivery-vehicle driving tasks, resulting in a drop in performance. In contrast, *Diffusion Planner* demonstrates strong transferability across different driving behaviors. We also compared works that utilize diffusion for planning, as shown in Table 4 in Appendix B.1. The *Diffusion Planner* better leverages the powerful capabilities of diffusion and is more practical.

**Qualitative Results**. To further demonstrate the capabilities of learning-based models, we show the trajectory generation results of representative baselines (without refinement) as shown in Figure 3. *Diffusion Planner* shows high-quality trajectory generation, with accurate predictions for neighboring vehicles and smooth ego planning trajectories that reasonably account for the speed of the vehicle ahead, demonstrating the advantages of joint modeling of both prediction and planning tasks. More closed-loop planning results are shown in Appendix A. In contrast, *GameFormer w/o refine* produces less smooth trajectories and inaccurate predictions for neighboring vehicles, which explains why it heavily relies on refinement. Although *PlanTF* and *PLUTO w/o refine.* sample multiple trajectories at once, most of them are of low quality.

### 5.1 EMPIRICAL STUDIES OF DIFFUSION PLANNER PROPERTIES

**Multi-modal Planning Behavior**. We selected an intersection scenario and performed multiple inferences without low temperature sampling from the same initial position to obtain different possible outputs, in order to evaluate the model's ability to fit multi-modal driving behaviors. As shown in Figure 5, without navigation information, the vehicle can exhibit three distinct driving behaviors—left turn, right turn, and straight ahead—with clear differentiation. When navigation information is provided, the model accurately follows it to make a left turn, demonstrating the diffusion model's ability to fit driving behaviors with varying distributions and its capacity for switching between them.

**Flexible guidance mechanism**. Based on the trained *Diffusion Planner* model, different types of classifier guidance, as described in Section 4.3, are added during inference time without requiring additional training. We present two cases to demonstrate the effectiveness of guidance and its flexible composability, as shown in Figure 4. 1)

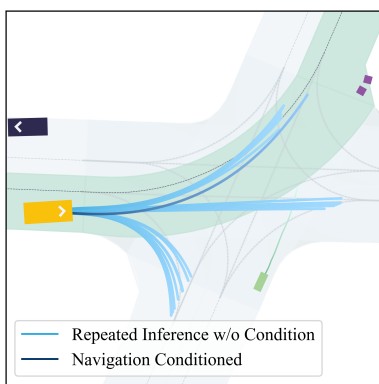

Figure 5: Multi-modal planning behavior of *Diffusion Planner*.

Table 3: Ablation of each modules during the training process on nuPlan Test14 Benchmark.

| Type | Planner | Score |
|------|---------|-------|
| Base | Diffusion Planner | 89.19 |
| Data | w/o z-score norm | 85.02 |
| | w/o interpolation | 83.78 |
| | w/o augmentation | 76.53 |
| Ego state | w/ SDE | 82.90 |
| | w/ ego state | 78.65 |
| | w/o current state | 81.11 |

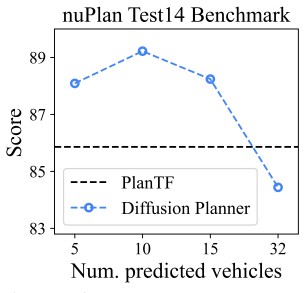

Figure 6: Ablation on the number of predicted vehicles $M$.

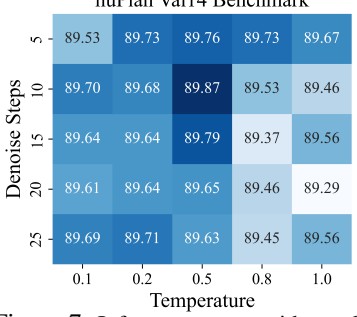

Figure 7: Inference param grid search.

For target speed setting, we masked all lane speed limit information to prevent it from influencing the model's planning, ensuring that speed adjustments are made solely through guidance. As a result, the model exhibited a lower speed without guidance. By setting the speed between 10m/s and 14m/s, the model closely matches the desired speed range while maintaining smooth speed transitions. 2) For comfort guidance, we effectively alleviate discomfort and can even use it simultaneously with collision guidance. We also provide additional case studies on collision and drivable guidance, as shown in Appendix B.2, as well as cases demonstrating the flexible combination of collision and drivable guidance, as illustrated in Figure 2.

## 5.2 ABLATION STUDIES

**Design Choices for training**. We demonstrate the effectiveness of key components of our method: data processing, the handling approach of ego current state, and the number of predicted vehicles. 1) We ablate the model's performance without using z-score normalization (*w/o z-score norm*), as well as without data augmentation (*w/o augmentation*), or by only perturbing the current state without applying interpolation to future trajectories (*w/o interpolation*). The results are summarized in Table 3. For the *w/o z-score norm* variant, even with ego-centric transformation, the data range remains large, making it difficult for the model to fit the distribution. The *w/o augmentation* variant faces out-of-distribution issues, leading to poor performance. Results also show that future trajectory interpolation is essential compared to perturbing only the current state. 2) We analyze the impact of the ego vehicle's current state on the model. Retaining velocity, acceleration, and yaw rate (*w/ ego state*) may lead to learning shortcuts, resulting in decreased planning capability. While a state dropout encoder (Cheng et al., 2023) (*w/ SDE*) mitigates this, directly discarding the information is more effective. Additionally, the *w/o current state* shows that adding current state information to the decoder improves planning capability. 3) We also ablate the choice of the number of $M$. Figure 6 shows that including too many neighboring vehicles in the decoder introduces noise, affecting the performance of the ego vehicle. However, most choices still outperform *PlanTF*.

**Design Choices for Inference**. We sweep two hyperparameters: the number of denoise steps and the magnitude of low-temperature sampling, as shown in Figure 7. Low temperature helps improve the stability of the output trajectories. Additionally, the model leverages DPM-Solver to achieve efficient denoising and remains robust across different step counts. We report the detailed parameter selection in Table 5.

## 6 CONCLUSION

We propose *Diffusion Planner*, a learning-based approach that fully exploits the expressive power and flexible guidance mechanism of diffusion models for high-quality autonomous planning. A transformer-based architecture is introduced to jointly model the multi-modal data distribution in motion prediction and planning tasks through a diffusion objective. Classifier guidance is employed to align planning behavior with safe or user preferred driving styles. *Diffusion Planner* achieves state-of-the-art closed-loop performance without relying on any rule-based refinement on the nuPlan benchmark and a newly collected 200-hour delivery-vehicle driving dataset, demonstrating strong adaptability across diverse driving styles. Due to space limit, more discussion on limitations and future direction can be found in Appendix E.

ACKNOWLEDGEMENT

This work is supported by National Key Research and Development Program of China under Grant (2022YFB2502904), and funding from Haomo.AI.

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

## A  VISUALIZATION OF CLOSED-LOOP PLANNING RESULTS

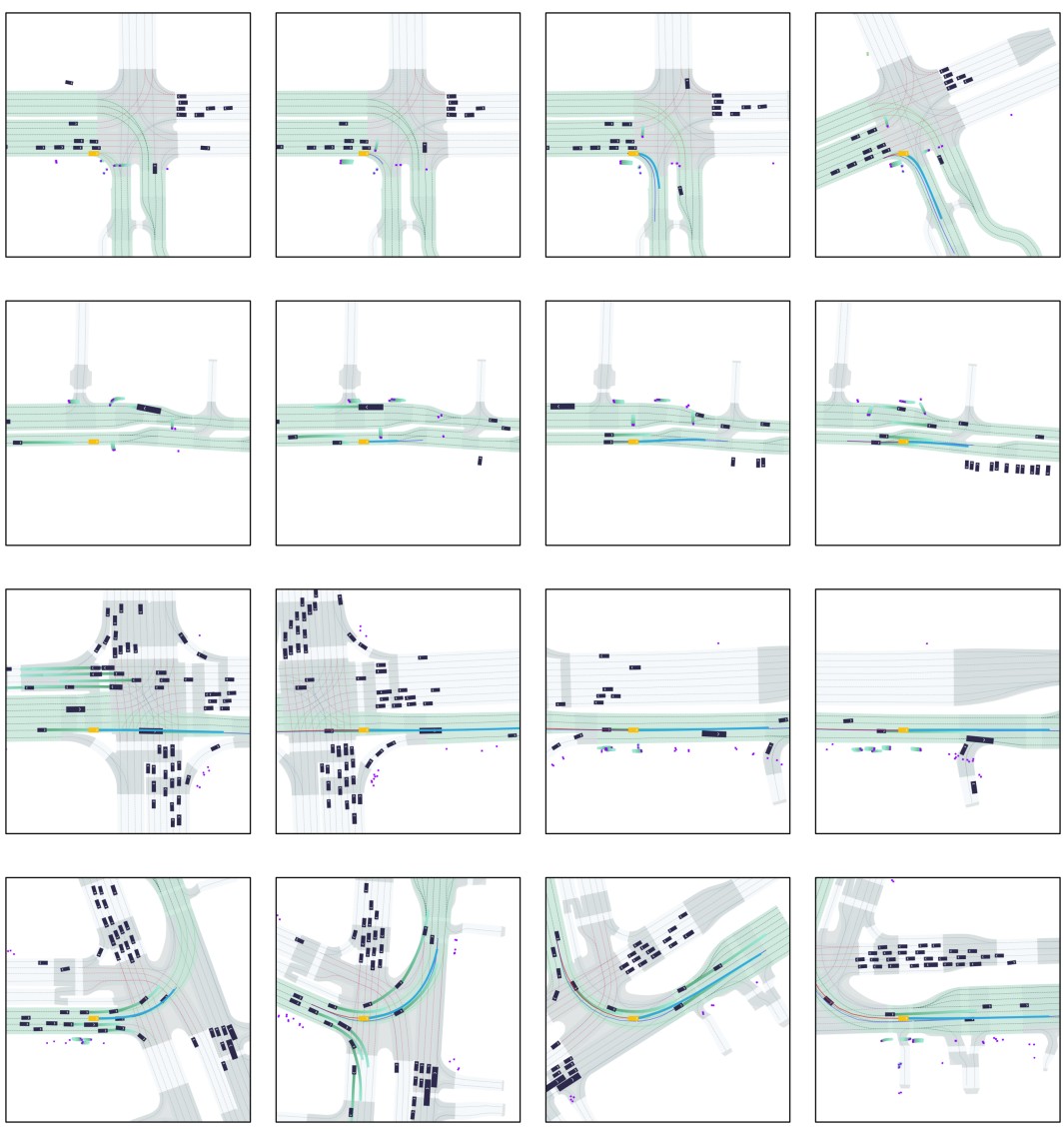

Figure 8: Closed-loop planning results: each row represents a scenario at 0, 5, 10, and 15 seconds intervals. Each frame includes the future planning of the ego vehicle, predictions for neighboring vehicles, the ground truth ego trajectory, and the driving history of the ego vehicle.

## B  ADDITIONAL RESULTS

### B.1  COMPARED TO DIFFUSION-BASED PLANNING METHODS

To further demonstrate the advantages of our model, we compared it with two recent works using diffusion models for motion planning. Diffusion-es (Yang et al., 2024) enhances a diffusion model by incorporating an LLM as a trajectory filter. STR-16M (Sun et al., 2023) uses a diffusion model as a decoder. STR2-CPKS-800M (Sun et al., 2024) builds on the former with 800M parameters and includes a PDM-like refinement module. We compared the model's performance in non-reactive mode and recorded the inference time, as shown in Table 4. We observe that current diffusion-based methods also experience significant performance degradation when detached from LLMs or rule-based refinement. Another important point is that these methods, due to their reliance on LLMs

or a large number of model parameters, have higher computational costs, making them difficult to deploy in real-world applications.

Table 4: Closed-loop non-reactive planning results on the nuPlan dataset among diffusion-based planners.

| Planner | Test14 | Test14-hard | Val14 | Inference Time (s) |
|---|---|---|---|---|
| Diffusion-es w/o LLM | - | - | 50 | - |
| Diffusion-es w/ LLM | - | - | 92 | 0.5 |
| STR-16M | - | 27.59 | 45.06 | - |
| STR2-CPKS-800M w/o refine. | 68.74 | 52.57 | 65.16 | >11 |
| Diffusion Planner (ours) | 89.19 | 75.99 | 89.87 | 0.04 |

### B.2 MORE CASE STUDIES FOR THE GUIDANCE MECHANISM.

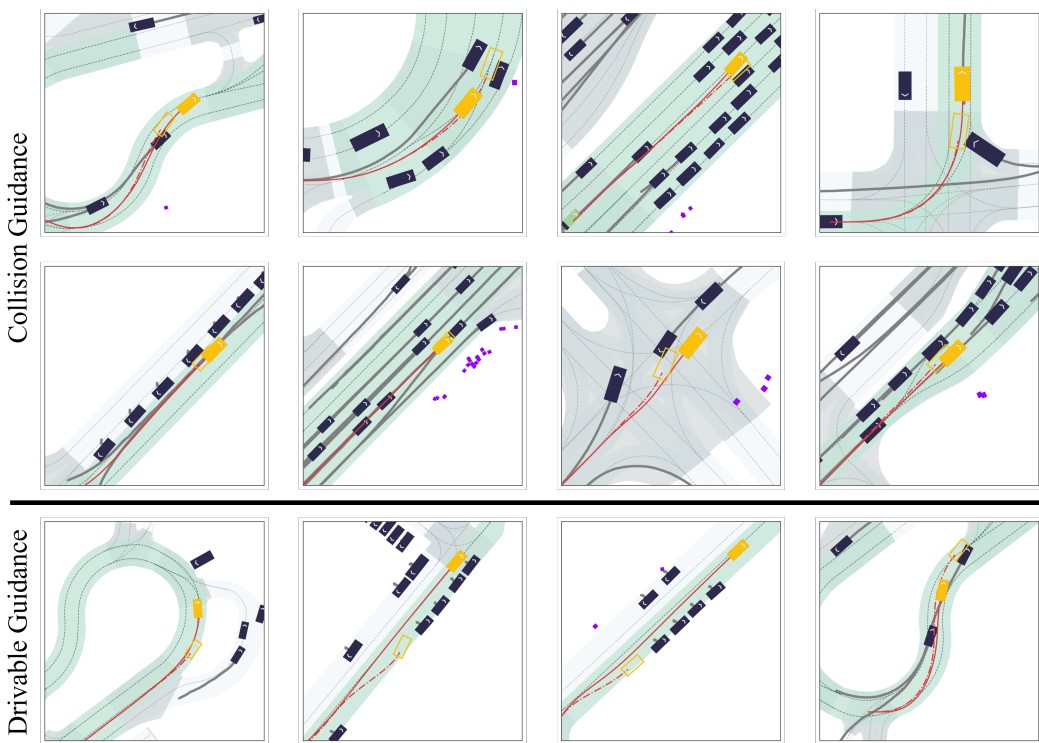

Figure 9: Case studies for collision and drivable guidance. Starting from the same position, we visualized the closed-loop test results: the dashed line represents the results without guidance, with hollow car markers indicating locations where safety incidents occurred. The solid line represents the results with guidance, and the solid car markers indicate the final positions.

## C EXPERIMENTAL DETAILS

This section outlines the experimental details to reproduce the main results in our papers.

### C.1 TRAINING DETAILS

**Datasets**. We use the training data from the nuPlan dataset and sample 1 million scenarios for our training set. The number of different scenarios is shown in Figure 11. For each scenario, we

https://www.nuplan.org/

consider the lane and navigation information within a $100m$ radius around the ego vehicle at the current time, including the neighboring vehicles' history from the past two seconds. Each type of data is padded to a unified dimension for model input, and attention masking is used to effectively eliminate irrelevant information.

**Data augmentation**. The current state of the ego vehicle is first perturbed slightly in terms of its $x$, $y$ coordinates, orientation angle $\theta$, speed $v$, acceleration $a$.

$$\Delta x^0 \sim \mathbb{U}\left([-\Delta x, -\Delta y, -\Delta \theta, -\Delta v, -\Delta a], [\Delta x, \Delta y, \Delta \theta, \Delta v, \Delta a]\right).$$

For the augmented state $\tilde{x}_{ego}^0 = x_{ego}^0 + \Delta x^0$, we ensure that the speed $v$ always remains greater than 0 to prevent the vehicle from learning to move in reverse. After that, a quintic polynomial interpolation is applied between current state $\tilde{x}_{ego}^0$ and $x_{ego}^{\tau 2s}$ to generate a new trajectory that adheres to the dynamic constraints, replacing the ground truth trajectory.

**Normalization**. Following previous works (Huang et al., 2023; Cheng et al., 2023; 2024), we apply an ego-centric transformation to process the original dataset. The global coordinates are converted into the ego vehicle's local coordinate system, using the vehicle's heading and position. Afterward, we observe that the ego vehicle's longitudinal progress is significantly larger than its lateral progress. To improve training stability, we apply z-score normalization to all x-axis coordinates, while the y-axis is scaled to the same magnitude to avoid distortion:

$$\tilde{x} = \frac{x - \mu}{\sigma}, \quad \tilde{y} = \frac{y}{\sigma},$$

where $\mu = 10$, $\sigma = 20$. The same approach is applied other scenario inputs.

Training was conducted using 8 NVIDIA A100 80GB GPUs, with a batch size of 2048 over 500 epochs, with a 5-epoch warmup phase. We use AdamW optimizer with a learning rate of $5e^{-4}$. We report the detailed setup in Table 5.

## C.2 INFERENCE DETAILS

We utilize DPM-Solver++ as diffusion reverse process solver, adopting variance-preserving(VP) noise schedule where the noise is $\sigma_t = (1-t)\beta_{\min} + t\beta_{\max}$. Low-temperature sampling is employed to further enhance the stability of the denoising process. We found that directly using the model output with a higher temperature facilitates generating high-quality trajectories. Conversely, if a refinement module is applied after the model output, a lower temperature helps produce more stable trajectories, which supports more accurate judgments by the refinement module. In addition, the model achieves an inference frequency of 20 Hz on a single A6000 GPU. We also report the detailed setup in Table 5.

## C.3 CLASSIFIER GUIDANCE DETAILS

We then specifically introduce the mathematical formulation of the different energy functions, as mentioned in Section 4.3.

**Collision Avoidance**. Based on the ego vehicle's planning and the neighboring vehicles' predictions from the decoder at diffusion timestamp $t$, we calculate the signed distance $\mathbf{D}$ between the ego vehicle and each neighboring vehicle at each timestamp $\tau$. When the bounding boxes of the vehicles overlap, we use the minimum separation distance, otherwise, we use the distance between the nearest points. The energy function for collision avoidance is then defined as:

$$\mathcal{E}_{\text{collision}} = \frac{1}{\omega_c} \cdot \frac{\sum_{M,\tau} \mathbb{1}_{\mathbf{D}_M^\tau > 0} \cdot \Psi\left(\omega_c \cdot \max\left(1 - \frac{\mathbf{D}_M^\tau}{r}, 0\right)\right)}{\sum_{M,\tau} \mathbb{1}_{\mathbf{D}_M^\tau > 0} + \text{eps}}$$
$$+ \frac{1}{\omega_c} \cdot \frac{\sum_{M,\tau} \mathbb{1}_{\mathbf{D}_M^\tau < 0} \cdot \Psi\left(\omega_c \cdot \max\left(1 - \frac{\mathbf{D}_M^\tau}{r}, 0\right)\right)}{\sum_{M,\tau} \mathbb{1}_{\mathbf{D}_M^\tau < 0} + \text{eps}},$$

(9)

where $\Psi(x) := e^x - x$, $r$ represents the collision-sensitive distance, which controls the maximum distance at which gradients are produced, and eps is added to ensure numerical stability (Jiang et al., 2023).

**Target Speed Maintenance**. We calculate the energy function based on the difference between the average speed of the generated trajectory and the target speed range:

$$\mathcal{E}_{\text{target\_speed}} = \max\left(\overline{\frac{\mathrm{d}x^\tau_{\text{ego}}}{\mathrm{d}\tau}} - v_{\text{low}}, 0\right)^2 + \max\left(v_{\text{high}} - \overline{\frac{\mathrm{d}x^\tau_{\text{ego}}}{\mathrm{d}\tau}}, 0\right)^2. \tag{10}$$

Where $v_{\text{low}}$ is the setting lower bound of speed, $v_{\text{low}}$ is the setting higher bound of speed.

**Comfort**. Taking longitudinal jerk as an example, the difference between each point and the comfort threshold is calculated, ignoring cases where the comfort requirements are met:

$$\mathcal{E}_{\text{comfort}} = \mathbb{E}\left[\max\left(\left(j_{\text{max}} - \left|\frac{\mathrm{d}^3 x^\tau_{\text{ego}}}{\mathrm{d}\tau^3}\right|\right)\Delta\tau^3, 0\right)^2\right]. \tag{11}$$

Where $j_{\text{max}}$ is the maximum longitude jerk limit.

**Staying within Drivable Area**. We construct the differentiable cost map $\mathbf{M}$ by using Euclidean Signed Distance Field with parallel computation (Cheng et al., 2024), which can compute the distance the ego vehicle goes beyond the lane at each timestamp. Then the energy is defined as:

$$\mathcal{E}_{\text{drivable}} = \frac{1}{\omega_{\text{d}}} \cdot \frac{\sum_\tau \Psi\left(\omega_{\text{d}} \cdot \mathbf{M}\left(x^\tau_{\text{ego}}\right)\right)}{\sum_\tau \mathbb{1}_{\mathbf{M}\left(x^\tau_{\text{ego}}\right)>0} + \text{eps}}. \tag{12}$$

Given the diverse options for energy function design, our choices were made primarily to validate whether the model could support various types of guidance and may not be optimal. However, through extensive empirical experiments, we can share some of our insights and experiences regarding energy function selection to assist future work in exploring more effective options:

- **Smooth and continuous gradients**: Guidance functions with smooth and continuous gradients facilitate the generation of stable trajectories.
- **Gradient sparsity**: It is preferable for the guidance function to generate gradients only in specific situations, such as when trajectory points approach potential collisions.
- **Indirect guidance for higher-order state derivatives**: For higher-order state derivatives, such as velocity, acceleration, or angular velocity, indirect guidance through position and heading is preferable. For instance, to control trajectory speed, we can guide trajectory length instead.
- **Consistent gradient magnitude**: The guidance function should ensure that the magnitude of gradients remains approximately consistent across different conditions. It can be achieved by averaging cost values over the number of points contributing to the cost.

### C.4 BASELINES SETUP

**nuPlan Datasets Evaluation**. For *IDM* and *UrbanDriver*, we use the official nuPlan code, with the *UrbanDriver* checkpoint sourced from the *PDM* codebase, which also provides the checkpoints for *PDM-Hybrid* and *PDM-Open*. For *PlanTF* and *PLUTO*, we use the checkpoints from their respective official codebases. In the case of *PLUTO w/o refine*, we skip the post-processing code and rerun the simulation without retraining. Following the guidelines from the official codebase, we train *GameFormer* and skip the refinement step to obtain *GameFormer w/o refine*.

**Delivery-vehicle Datasets Evaluation**. We adopt the same metrics and models as those used on nuPlan, but by modifying various vehicle-related parameters to adapt the baselines to the delivery-vehicle training. Based on this, we retrain and test the models following the official training code.

https://github.com/motional/nuplan-devkit
https://github.com/autonomousvision/tuplan_garage
https://github.com/jchengai/planTF
https://github.com/jchengai/pluto
https://github.com/MCZhi/GameFormer-Planner

Table 5: Hyperparameters of *Diffusion Planner*

| Type | Parameter | Symbol | Value |
|---|---|---|---|
| Training | Num. neighboring vehicles | - | 32 |
| | Num. past timestamps | $L$ | 21 |
| | Dim. neighboring vehicles | $D_{\text{neighbor}}$ | 11 |
| | Num. lanes | - | 70 |
| | Num. points per polyline | $P$ | 20 |
| | Dim. lanes vehicles | $D_{\text{lane}}$ | 12 |
| | Num. navigation lanes | $D$ | 25 |
| | Num. predicted neighboring vehicles | $M$ | 10 |
| | Num. encoder/decoder block | - | 3 |
| | Dim. hidden layer | - | 192 |
| | Num. multi-head | - | 6 |
| Inference | Noise schedule | - | Linear |
| | Noise coefficient | $\beta_{\min}, \beta_{\max}$ | 0.1, 20.0 |
| | Temperature | - | 0.5 |
| | Temperature (w/ refine.) | - | 0.1 |
| | Denoise step | - | 10 |

## D    DETAILS ON DELIVERY VEHICLE EXPERIMENTS

We collected approximately 200 hours of real-world data using an autonomous logistics delivery vehicle from Haomo.AI. The task of the delivery vehicle is similar to that of a robotaxi in nuPlan, as it autonomously navigates a designated route. During operation, the vehicle must comply with traffic regulations, ensure safety, and complete the delivery as efficiently as possible. Compared to the vehicles in the nuPlan dataset, the delivery vehicle is smaller, as shown in Table 6, and operates at lower speeds. As a result, it is able to travel on both main roads and bike lanes. During deliveries, it frequently interacts with pedestrians and cyclists, and the driving rules differ from those for motor vehicles, as shown in 10. This dataset serves as a supplement to nuPlan, allowing for the evaluation of algorithm performance under diverse driving scenarios.

Table 6: Vehicle parameter details

| Parameter (m) | Delivery Vehicle | nuPlan Vehicle |
|---|---|---|
| Width | 1.03 | 2.30 |
| Length | 2.34 | 5.18 |
| Height | 1.65 | 1.78 |
| Wheel base | 1.20 | 3.09 |

Specifically, we transform the original data into the nuPlan data structure, allowing it to be stored as DB files compatible with the nuPlan API for seamless integration and usage. We use the same training pipeline from the nuPlan benchmark to train both the model and baselines. For some baselines that require crosswalk information, we replace it with stop line data. Additionally, the vehicle parameters are substituted with those of the delivery vehicle. The model's performance is evaluated using the nuPlan metrics.

## E    LIMITATIONS & DISCUSSIONS & FUTURE WORK

Here, we discuss our limitations, potential solutions and interesting future works.

- **Scenario Inputs**. Our method relies on vectorized map information and detection results of neighboring vehicles. Compared to mainstream end-to-end pipelines, this approach involves some information loss and requires a data processing module. However, unlike end-to-end methods, our focus is more on the planning stage, particularly on the ability for closed-loop planning.

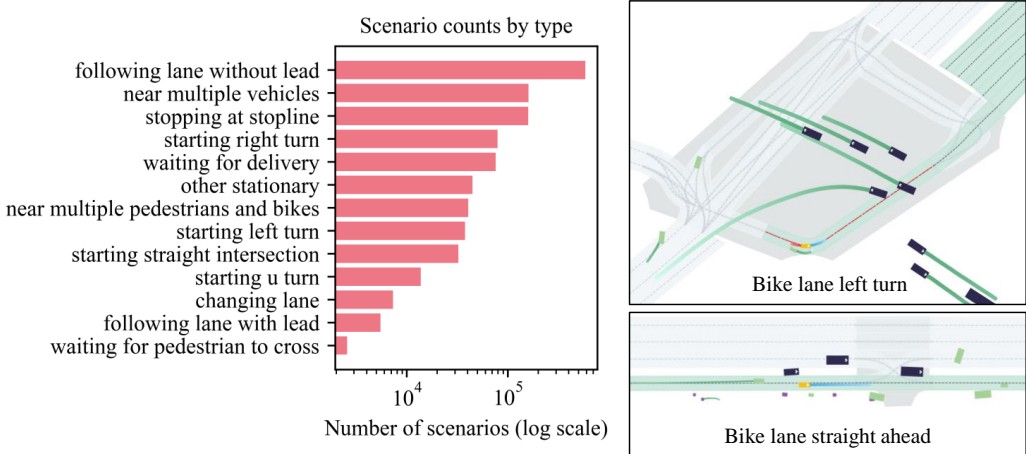

Figure 10: Scenario count by type in the delivery-vehicle driving dataset, with representative visualizations.

*Solution and future work:* We demonstrate the performance of the diffusion model for closed-loop planning without rule-based refinement. An interesting future direction would be to modify the encoder architecture and use images as inputs, enabling an end-to-end training pipeline.

- **Lateral Flexibility**. We find that learning-based methods struggle with flexibility, particularly when significant lateral movement is required. In contrast, rule-based methods perform better in this aspect due to the provision of a reference trajectory. Being consistent with findings from previous work (Li et al., 2024), we find this is mostly because that the dataset mainly consists of straight-driving scenarios, with few instances of lane changes or avoidance maneuvers. This makes it challenging for learning-based methods to generalize and acquire these skills. Additionally, since the model only outputs the planned trajectories instead of the controlling signal such as brake and throttle, there is a gap between the planned trajectory and the results from the downstream controller (Cheng et al., 2023). This discrepancy also leads to potential poor performance, or even out-of-distribution behavior, in scenarios that require more flexible actions.

  *Solution and future work:* We find that data augmentation can somewhat alleviate the issue of the vehicle being reluctant to make lateral movements, but it still performs poorly in cases requiring significant lane changes. This could be improved by incorporating more data involving large lateral progress, leveraging reinforcement learning with a reward mechanism, or designing a more effective diffusion guidance mechanism to help the model learn lane-changing behaviors. We believe this is an interesting observation and leave this direction for future work.

- **Sample Efficiency**. The high performance of Diffusion comes at the cost of requiring multiple model inferences, leading to reduced sample efficiency.

  *Solution and future work:* We addressed this issue to a large extent by using a high-order ODE solver, enabling trajectory planning for 8 seconds at 10 Hz in 0.05 seconds. Considering real-world application requirements, techniques such as consistency models (Song et al., 2023) or distillation-based sampling methods (Meng et al., 2023) could be employed for further acceleration.

Overall, although some design choices may appear simple and certain limitations exist, we have thoroughly demonstrated the capabilities of diffusion models for closed-loop planning in autonomous driving through extensive experiments. Moreover, we demonstrate the potential of the diffusion model to align with safety or human-preferred driving behaviors. It provides a high-performance, highly adaptable planner for autonomous driving systems.

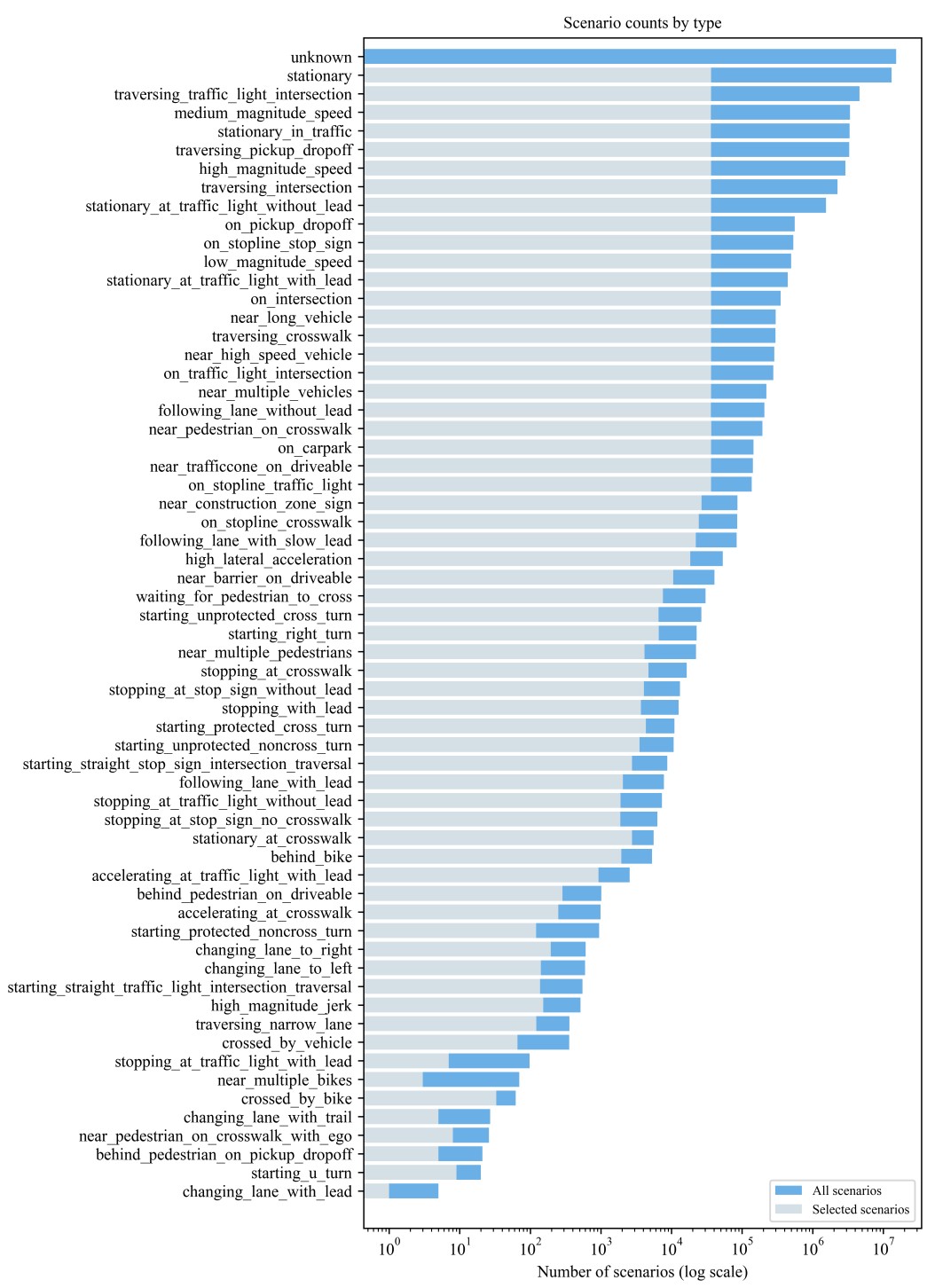

Figure 11: Scenario count by type in the nuPlan dataset.

