# OpenReview forum: "Diffusion-Based Planning for Autonomous Driving with Flexible Guidance"
_ICLR.cc/2025/Conference — ICLR 2025 Oral_

### Official Review · Reviewer_d65s · 2024-10-30

**Soundness:** 3
**Presentation:** 2
**Contribution:** 3
**Rating:** 8
**Confidence:** 3

**Summary:**

This paper proposes a diffusion-based motion planner for autonomous driving. The diffusion model is trained to generate joint trajectories of the ego and neighboring vehicles. Classifier guidance is introduced to regularize the safety, comfort, and feasibility of the ego vehicle's trajectory. Several practical implementation designs, such as data augmentation and normalization, are introduced to improve the model performance for closed-loop planning. Comprehensive experiments are conducted on the nuPlan benchmark.

**Strengths:**

The proposed model is carefully designed and extensively tailored for practical application in autonomous driving. The experiments are comprehensive, with multiple baselines and detailed ablation studies. It achieves better performance than the current state-of-the-art methods on the nuPlan leaderboard. Detailed descriptions of the model implementation and experiment design are provided. This work provides useful lessons for the community on achieving good closed-loop planning performance with diffusion-based motion planners for practical autonomous driving applications.

**Weaknesses:**

1. While I believe this work has good practical values for the autonomous driving community, diffusion models have been explored in several works for motion prediction, closed-loop simulation, and motion planning in autonomous driving. The authors attempted to differentiate their work from existing works by emphasizing that the diffusion model does not drive performance improvements in these cases but relies on rule-based refinement or LLM, and they claimed that they are the first to fully harness the potential of diffusion models to enhance closed-loop planning performance in autonomous driving. However, this statement is quite vague, and it is unclear what novel technical contributions the authors have made in this work. My current impression is that the proposed model combines existing techniques investigated in the literature on diffusion models and learning-based motion planning for autonomous driving (e.g., transformer architecture, classifier guidance with manually designed cost functions, data augmentation, etc.). To help the audience better appreciate their novelty and contributions, the authors may list their novel contributions at the end of the Introduction.

2. Related to the first point, the authors should provide a more extensive review of the related literature on diffusion models for autonomous driving. The authors currently give a concise discussion in Sec. 3.2. It should be moved to the related work section and extended to be more comprehensive. In particular, the current review misses an essential line of research on diffusion models for closed-loop simulation (e.g., [1], [2], [3]). Also, some works have developed diffusion-based motion planners for nuPlan (e.g., [4], [5], and [6]; the authors cited [4][5], but they are not included as baselines). If possible, they should be included as baselines for comparison.

[1] Zhong, Ziyuan, et al. "Guided conditional diffusion for controllable traffic simulation." ICRA 2023.
[2] Zhong, Ziyuan, et al. "Language-guided traffic simulation via scene-level diffusion." CoRL 2023.
[3] Chang, Wei-Jer, et al. "Safe-sim: Safety-critical closed-loop traffic simulation with diffusion-controllable adversaries." ECCV 2024.
[4] Yang, Brian, et al. "Diffusion-es: Gradient-free planning with diffusion for autonomous driving and zero-shot instruction following." arXiv preprint arXiv:2402.06559 (2024).
[5] Hu, Yihan, et al. "Solving motion planning tasks with a scalable generative model." ECCV 2024.
[6] Sun, Qiao, et al. "Large Trajectory Models are Scalable Motion Predictors and Planners." arXiv preprint arXiv:2310.19620 (2023).

**Questions:**

1. In Table 1, the authors only reported the proposed diffusion planner with collision guidance. How does the model perform with other types of guidance, especially when multiple guidance costs are used?

2. When multiple guidance terms are used, the guidance cost becomes a weighted sum of several cost terms. I wonder if the authors have taken special care to ensure that the scales of different cost terms and their gradients are well balanced. It would be great if the authors could discuss if the guidance performance is sensitive to energy function design and cost weights.

---

> ### Author Response · Authors · 2024-11-19
> **Response to Reviewer d65s 1/2**
>
> We thank the reviewer for the constructive comments. Regarding the concerns of the reviewer d65s, we provide the following responses.
>
> > **W1. It is unclear what novel technical contributions the authors have made in this work.**
>
> We appreciate the reviewer for the constructive comment. We have summarized our contributions at the end of the introduction in the revised version. We provide the discussion as follows:
>
> - To the best of the authors' knowledge, this paper is the first to fully harness the power of diffusion models for high-performance motion planning, without reliant on rule-based refinement. This is further demonstrated in the General Response GR1, where we test our model on larger and more challenging benchmarks, still achieving SOTA performance.
> - We designed a model specifically for planning, not just combining existing techniques, achieving high-quality trajectory outputs and SOTA performance (Figure 3, General Response GR1). Our DiT-based architecture features a core fusion mechanism between noised future vehicle trajectories and conditional inputs, such as environmental and navigation information. We redefine planning as a future trajectory generation task, jointly producing the ego vehicle's plan and neighboring vehicle predictions. Additionally, we show how the guidance mechanism in diffusion models can align planning behavior with safe or human-preferred driving styles.
> - We have collected and evaluated a new 200-hour delivery-vehicle dataset, carefully processing it to ensure compatibility with the nuPlan framework. We will open-source this dataset as a supplement to nuPlan.
>
> > **W2. The authors should provide a more extensive review of the related literature on diffusion models for autonomous driving.**
>
> We thank the reviewer for the suggestion to discuss related works. We follow the recommendation to moved the discussion on diffusion-based methods to the "Related Works" section and have also expanded the discussion on traffic simulation-related works, which is briefly outlined below.
>
> - For [1], [2], and [3] ([1],[2],[3] in the reviewer's comment), which focus on generating diverse driving scenarios, it is important to note that this task differs from planning. Their generation emphasizes diversity in simulation rather than quality or drivability, as the outputs are not directly used for control. Additionally, their method uses the standard UNet structure from diffuser [4] without design considerations specific to autonomous driving planning.
>
> > **W3. Some works have developed diffusion-based motion planners for nuPlan, they should be included as baselines for comparison .**
>
> We thank the reviewer for recommending related works [5] [6] [7] ([4],[5],[6] in the reviewer's comment) as baselines. During the rebuttal phase, **we compared them, and our method continued to achieve superior performance (see Table below or more details in General Response GR1)**. These methods naively apply diffusion loss to existing frameworks or stack parameters without specific designs, making them reliant on post-processing and resulting in high computational costs. Detailed implementation is provided below:
>
> - For [5], we use the Val14 scores for the diffusion policy under the no-LLM condition reported in the paper, as well as the inference time with LLM assistance.
> - For [6], this is not a diffusion-based method. While the official checkpoint is provided, the code for closed-loop simulations is not available, making it unusable for our comparison. Moreover, the paper only reports open-loop scores and scores under the reactive setting with refinement, which limits our ability to make a meaningful comparison.
> - For [7], the implementation is only available in an open-loop setting. However, we found the corresponding closed-loop scores in its v2 version [8] (released after our submission). Additionally, we used the publicly available 800M checkpoint from [8], removed the rule-based refinement, and re-tested the model, also evaluating the inference time.
>
> | Methods                         | Test14 | Test14-hard | Val14 | Inference Time (s) |
> | ------------------------------- | ------ | ----------- | ----- | ------------------ |
> | [5] w/o LLM                     | -      | -           | 50    | -                  |
> | [5] w/ LLM                      | -      | -           | 92    | 0.5                |
> | [7] (Not sure if it has refine) | -      | 27.59       | 45.06 | -                  |
> | [8] w/o refine.                 | 14.77  | 10.99       | 8.80  | >11                |
> | Diffusion Planner (ours)        | 89.22  | 75.67       | 89.76 | 0.08               |

---

> > ### Author Response · Authors · 2024-11-19
> > **Response to Reviewer d65s 2/2**
> >
> > > **Q1. How does the model perform with other types of guidance, especially when multiple guidance costs are used ?**
> >
> > - We have added the combination of collision and comfort guidance in Figure 4. Since we aim to compare our performance with other learning-based algorithms, introducing extensive guidance designs could lead to unfair comparisons. Additionally, some types of guidance may not be fully reflected in the scores. For example, guiding based on user-set speeds might deviate from expert trajectories, resulting in lower scores in the nuPlan evaluation. We primarily showcase the feasibility of the guidance mechanism and leave the design of detailed guidance energy functions as a user choice for practical implementation.
> > - To further address the reviewer's concern, we have included drivable guidance alongside collision guidance to demonstrate the effect of multiple guidance types on closed-loop simulation, as shown below.
> > - We have also added a case study visualization in Appendix F of the revised version, which includes an example where the ego vehicle, under collision guidance alone, veers off the road to avoid a rear-approaching vehicle. With the addition of drivable guidance, the vehicle remains on the road while maintaining safety.
> >
> > | Diffusion Planner | w/ Collision Guidance | w/ Collision & Drivable Guidance |
> > | ----------------- | --------------------- | -------------------------------- |
> > | 89.22             | 91.05                 | 92.13                            |
> >
> > > **Q2. It would be great if the authors could discuss if the guidance performance is sensitive to energy function design and cost weights.**
> >
> >
> > Given the diverse options for energy function design, our choices were primarily made to validate whether the model could support various types of guidance and may not be optimal. However, through extensive empirical experiments, we can share some of our insights and experiences regarding energy function selection to assist future work in exploring more effective options. Relevant content has been added in Appendix C.3 of the revised version.
> >
> > -  **Smooth and continuous gradients**: Guidance functions with smooth and continuous gradients facilitate the generation of stable trajectories.
> > -  **Gradient sparsity**: It is preferable for the guidance function to generate gradients only in specific situations, such as when trajectory points approach potential collisions.
> > -  **Indirect guidance for higher-order state derivatives**: For higher-order state derivatives, such as velocity, acceleration, or angular velocity, indirect guidance through position and heading is preferable. For instance, to control trajectory speed, we can guide trajectory length instead.
> > -  **Consistent gradient magnitude**: The guidance function should ensure that the magnitude of gradients remains approximately consistent across different conditions. It can be achieved by averaging cost values over the number of points contributing to the cost.
> >
> > The sensitivity of cost weights is closely related to the specific form of the cost function. For the energy functions we selected, we varied the guidance scale for collision and drivable guidance, as shown below. Our guidance demonstrated robust performance across different weights, outperforming the version without guidance.
> >
> > | Collision \ Drivable | 0.01  | 0.05  | 0.1   | 0.3   | 0.5   |
> > |-------|-------|-------|-------|-------|-------|
> > | **2.5**   | 89.99 | 89.61 | 91.22 | 90.74 | 90.50 |
> > | **3**     | 90.77 | 91.17 | 92.13 | 91.16 | 90.26 |
> > | **3.5**   | 90.96 | 90.74 | 91.68 | 91.15 | 90.80 |
> >
> >
> > [1] Zhong, Ziyuan, et al. "Guided conditional diffusion for controllable traffic simulation." 2023 IEEE International Conference on Robotics and Automation (ICRA). IEEE, 2023.
> >
> > [2] Zhong, Ziyuan, et al. "Language-guided traffic simulation via scene-level diffusion." *Conference on Robot Learning*. PMLR, 2023.
> >
> > [3] Chang, Wei-Jer, et al. "SAFE-SIM: Safety-Critical Closed-Loop Traffic Simulation with Diffusion-Controllable Adversaries." *European Conference on Computer Vision*. Springer, Cham, 2025.
> >
> > [4] Janner, Michael, et al. "Planning with diffusion for flexible behavior synthesis." arXiv preprint arXiv:2205.09991 (2022).
> >
> > [5] Yang, Brian, et al. "Diffusion-es: Gradient-free planning with diffusion for autonomous driving and zero-shot instruction following." *arXiv preprint arXiv:2402.06559* (2024).
> >
> > [6] Hu, Yihan, et al. "Solving motion planning tasks with a scalable generative model." *European Conference on Computer Vision*. Springer, Cham, 2025.
> >
> > [7] Sun, Qiao, et al. "Large Trajectory Models are Scalable Motion Predictors and Planners." *arXiv preprint arXiv:2310.19620* (2023).
> >
> > [8] Sun, Qiao, et al. "Generalizing Motion Planners with Mixture of Experts for Autonomous Driving." *arXiv preprint arXiv:2410.15774* (2024).

---

> > > ### Comment · Reviewer_d65s · 2024-11-22
> > >
> > > Thanks to the authors for the detailed response and additional experimental results. The response has addressed most of my questions and concerns. One comment remained:
> > >
> > > "For [1], [2], and [3] ([1],[2],[3] in the reviewer's comment), which focus on generating diverse driving scenarios, it is important to note that this task differs from planning. Their generation emphasizes diversity in simulation rather than quality or drivability, as the outputs are not directly used for control."
> > >
> > > In addition to diversity, realism is also a criterion considered in simulation and behavior generation. Collision and off-road rates, which quantify the quality or drivability of planning, are commonly used metrics for quantifying realism. In many of these works, the simulation model is also trained as a future trajectory generation task. Could the authors clarify what makes diffusion-based planning different from diffusion-based simulation or behavior generation? Since these two problems can both be solved as a trajectory generation task, the techniques developed for one problem could be applicable and beneficial for another. I don't think the overlap between these two problems diminishes the quality of the paper, given the solid and outstanding planning performance on challenging benchmarks and extensive experiments. Still, the connection between these two lines of work should be discussed appropriately (e.g., when defining the task in Sec. 4.1) to position the proposed work in the autonomous driving literature without overclaiming the novelty.

---

> > > > ### Author Response · Authors · 2024-11-22
> > > > **Thanks for the additional comments**
> > > >
> > > > We thank the reviewer for the detailed additional comments. Due to space limit, we will integrate the new results in the final version and conduct analysis and discussion in appropriate sections, such as Related Work and Section 4.1.
> > > >
> > > > We acknowledge that diffusion-based planning and diffusion-based simulation share similarities, as both focus on trajectory generation for autonomous driving and use diffusion loss during training. However, their objectives differ: **planning aims for a deployable, high-performance policy**, while **simulation emphasizes generating diverse and realistic driving scenarios**. This leads to differences in their future trajectory generation design:
> > > >
> > > > - **In diffusion-based simulation**, all traffic participants are often treated uniformly. Studies like [1] and [3] adopt UNet-based architectures, generating a single trajectory for each vehicle independently. On this basis, [2] proposed a scene-level diffusion model to simultaneously generate future trajectories for all participants. However, for each agent, independent agent-centric coordinate transformations are applied, necessitating additional modules to account for inter-agent relationships. Each agent also requires further fusion with its respective map information.
> > > > - **In contrast, diffusion planner** takes a different approach to future trajectory generation, with the focus remaining on the ego vehicle:
> > > >   - We also generate multiple vehicles' future trajectories simultaneously, but all information is represented in an ego-centric format. This eliminates the need for extra designs in [2].
> > > >   - Generating trajectories for too many neighboring vehicles negatively impacts ego performance, as shown in Figure 6. Therefore, we limit this to nearby vehicles, unlike simulation, which generates trajectories for all participants.
> > > >   - Unlike [2], we pre-fuse map and neighboring historical information, enhancing the ego vehicle’s scene understanding and significantly reducing computational costs during denoising. This makes the model better suited for high-frequency planning.
> > > >   - Compared to [1], [2], and [3], we implemented a more efficient navigation information fusion, removed the reliance on ego history information, and adopted augmentation strategies. All these design choices are aimed at enhancing the ego vehicle's closed-loop planning performance.
> > > > - In the definition of future trajectory generation in [1], [2], and [3], the models generate future actions and use a simplified vehicle model to compute states, supervising both states and actions simultaneously. In contrast, we directly supervise states, which is more straightforward and avoids biases introduced by the simplified dynamics.
> > > >
> > > > Finally, we want to discuss the difference between **planning drivability** and **simulation realism** in evaluation.
> > > >
> > > > - In **planning**, evaluating whether a trajectory is drivable involves integrating it with a downstream controller for closed-loop simulation, aligning with real-world application scenarios. If the generated trajectory is infeasible, the issue becomes exacerbated in closed-loop simulation, leading to out-of-distribution errors such as collisions or boundary violations, resulting in poor scores.
> > > >
> > > > - In contrast, **simulation** directly outputs actions, and realism is typically assessed based on certain metrics for these generated actions. A simplified bicycle model is often used for closed-loop simulation. While these actions may align well with data distributions and appear realistic, there is no evaluation of whether they can actually be executed by a controller.
> > > >
> > > > Compared to simulation realism, planning drivability provides a more practical and application-oriented assessment of driving strategies in real-world scenarios.
> > > >
> > > >
> > > >
> > > > [1] Zhong, Ziyuan, et al. "Guided conditional diffusion for controllable traffic simulation." 2023 IEEE International Conference on Robotics and Automation (ICRA). IEEE, 2023.
> > > >
> > > > [2] Zhong, Ziyuan, et al. "Language-guided traffic simulation via scene-level diffusion." *Conference on Robot Learning*. PMLR, 2023.
> > > >
> > > > [3] Chang, Wei-Jer, et al. "SAFE-SIM: Safety-Critical Closed-Loop Traffic Simulation with Diffusion-Controllable Adversaries." *European Conference on Computer Vision*. Springer, Cham, 2025.

---

> > > > > ### Comment · Reviewer_d65s · 2024-11-22
> > > > >
> > > > > Thank the authors for the detailed follow-up. The paper would benefit from adding a concise version of this discussion on the connection and difference between diffusion-based planning and simulation in the final version. Now I am satisfied with the paper  and have raised my score.

---

> > > > > > ### Author Response · Authors · 2024-11-23
> > > > > > **Thanks for increasing to 8!**
> > > > > >
> > > > > > Thank you so much for your effort engaged in the review phase and for increasing to 8! We’ll incorporate the discussion into the final version. Thanks again!

---

### Official Review · Reviewer_hoYE · 2024-11-02

**Soundness:** 3
**Presentation:** 3
**Contribution:** 3
**Rating:** 8
**Confidence:** 3

**Summary:**

In this paper, a DiT-enabled framework is deployed to jointly tackle the motion planning task accompanying with predictions of surrounding agents for autonomous driving. Subsequently, a classifier-free gradient guidance are leveraged through a set of safety and comfortness costs during the diffusion step in guiding safe planning. Experimental results in Test14 demonstrate the effectiveness.

**Strengths:**

1. A novel diffusion-based framework in solving the motion planning task. Intutive DiT-enabled framework for integrated prediction and planning with costs guidance.

2. Strong planning results delivered against state-of-the-art baselines in nuPlan.

**Weaknesses:**

1. Insufficient benchmark and metric comparison: 1) Additional results in other popular benchmark, such as Val14 and Test14-Hard are required to manifest the planning results under more diversed / challenging scenarios. 2) Other settings, such as closed loop reactive simulation, and open loop results are not verified.

2. Insufficient baseline comparison. Motion planning / trajectory simulation leveraging diffuision are not novel. Hence, the planning results using other diffusion strategies ([1] [2] for instance) seems needed.

Ref:

[1] Chi, Cheng, et al. "Diffusion policy: Visuomotor policy learning via action diffusion." The International Journal of Robotics Research (2023): 02783649241273668.

[2] Zhong, Ziyuan, et al. "Guided conditional diffusion for controllable traffic simulation." 2023 IEEE International Conference on Robotics and Automation (ICRA). IEEE, 2023.

**Questions:**

N/A.

---

> ### Author Response · Authors · 2024-11-19
> **Response to Reviewer hoYE 1/2**
>
> We thank the reviewer for the constructive comments and positive feedback on our paper. Regarding the concerns of the reviewer hoYE, we provide the following responses.
>
> > **W1. Insufficient benchmark and metric comparison**
>
> Regarding the comment on the Val14 and Test14-Hard results:
>
> - We appreciate the reviewer's constructive comment. We have added experimental results on the Val14 and Test14-Hard benchmarks. Please refer to General Response or Appendix B for more details. **We continue to achieve SOTA closed-loop performance among learning-based methods**.
>
> Regarding the comment on the open-loop and reactive results:
>
> - In our original submission, we primarily evaluated performance using closed-loop non-reactive simulation, as recent findings have shown that there is little correlation between open-loop prediction performance and closed-loop planning in both autonomous driving [1] [2] and robotics [3] [4]. Methods optimized for open-loop metrics can face severe distribution shift problems in closed-loop settings. For instance, [1] highlights the misalignment between ego-forecasting (open-loop) and planning (closed-loop), while [4] states that "action MSE on the validation set is another potential metric, but we find it often does not correlate with real-world performance." Due to the unreliability of open-loop evaluations, we chose to report the more reliable closed-loop performance. However, to fully address the reviewer's concern, we have included the open-loop results, as shown in the table below, which align with the above analysis.
> -  For the reactive metric, nuPlan employs a simple IDM model for neighboring vehicles, which is a relatively weak model and does not accurately reflect real-world traffic characteristics [5]. During the rebuttal phase, we conduct additional experiments in reactive settings, as shown below, and **our method continue to achieve sota performance among learning-based baselines**.
>
> | Methods                  | Val14 (non-reactive) | Val14 (reactive) | Val14 (open-loop) | Test14 (non-reactive) | Test14 (reactive) | Test14 (open-loop) | Test14-hard (non-reactive) | Test14-hard (reactive) | Test14-hard (open-loop) |
> | ------------------------ | -------------------- | ---------------- | ----------------- | --------------------- | ----------------- | ------------------ | -------------------------- | ---------------------- | ----------------------- |
> | PDM-Open*                | 53.53                | 54.24            | 85.88             | 52.81                 | 57.23             | 84.13              | 33.51                      | 35.83                  | 79.06                   |
> | UrbanDriver              | 68.57                | 64.11            | 84.21             | 51.83                 | 67.15             | 81.21              | 50.40                      | 49.95                  | 77.18                   |
> | GameFormer w/o refine.   | 13.32                | 8.69             | 70.3              | 11.36                 | 9.31              | 68.42              | 7.08                       | 6.69                   | 61.71                   |
> | PlanTF                   | 84.27                | 61.61            | 88.69             | 85.62                 | 79.58             | 87.80          | 69.70                      | 61.61                  | 84.86               |
> | PLUTO w/o refine.*       | 88.89                | 78.11            | 89.29         | **89.90**             | 78.62             | 87.81         | 70.03                      | 59.74                  | 81.40                   |
> | Diffusion Planner (ours) | **89.76**            | **82.56**        | 76.17             | **89.22**                 | **83.36**         | 72.99              | **75.67**                  | **68.56**              | 66.22                   |

---

> > ### Author Response · Authors · 2024-11-19
> > **Response to Reviewer hoYE 2/2**
> >
> > > **W2. Insufficient baseline comparison.**
> >
> > We appreciate the reviewer for suggesting these relevant works. We briefly discuss these methods and explain why they were not chosen as baselines:
> >
> > - For [3] ([1] in the reviewer's comment), which focuses on robotics manipulation tasks rather than autonomous driving, the method uses images as input and outputs short sequences (4 steps) of robotic arm torques. In contrast, our approach uses vector-based inputs to produce semantically meaningful trajectory outputs (80 steps), requiring strong long-sequence modeling capabilities. Additionally, while [3] incorporates ego history data for better performance, this can lead to models in autonomous driving replicating past behavior [6], which degrades closed-loop performance. These factors make [3] difficult to apply directly to autonomous driving.
> > - For [7] ([2] in the reviewer's comment), which focuses on generating diverse driving scenarios, it is important to note that this task differs from planning. Their generation emphasizes diversity in simulation rather than quality or drivability, as the outputs are not directly used for control. Additionally, their method uses the standard UNet structure from diffuser [8] without design considerations specific to autonomous driving planning.
> >
> >
> > Although many works use diffusion models for decision-making, **achieving superior closed-loop autonomous planning requires specific designs**.
> > - To further address the reviewer's concerns, we included two more relevant diffusion-based planning [9] [10] comparisons in our revised version (see General Response GR1 and Appendix B). Our method continues to achieve much better performance. These methods naively apply diffusion loss to existing frameworks or stack parameters without specific designs. Hence, they heavily rely on post-processing for reasonable performance. On the contrary, our method emphasizes interaction between the ego vehicle and surrounding vehicles, incorporates a diffusion transformer for better performance and scalability, and uses data augmentation to enhance model performance, along with ODE-accelerated sampling for deployability.
> >
> >
> > [1] Dauner, Daniel, et al. "Parting with misconceptions about learning-based vehicle motion planning." *Conference on Robot Learning*. PMLR, 2023.
> >
> > [2] Caesar, Holger, et al. "nuplan: A closed-loop ml-based planning benchmark for autonomous vehicles." *arXiv preprint arXiv:2106.11810* (2021).
> >
> > [3] Chi, Cheng, et al. "Diffusion policy: Visuomotor policy learning via action diffusion." The International Journal of Robotics Research (2023): 02783649241273668.
> >
> > [4] Lin, Fanqi, et al. "Data Scaling Laws in Imitation Learning for Robotic Manipulation." arXiv preprint arXiv:2410.18647 (2024).
> >
> > [5] Chang, Wei-Jer, et al. "SAFE-SIM: Safety-Critical Closed-Loop Traffic Simulation with Diffusion-Controllable Adversaries." *European Conference on Computer Vision*. Springer, Cham, 2025.
> >
> > [6] Cheng, Jie, et al. "Rethinking imitation-based planners for autonomous driving." *2024 IEEE International Conference on Robotics and Automation (ICRA)*. IEEE, 2024.
> >
> > [7] Zhong, Ziyuan, et al. "Guided conditional diffusion for controllable traffic simulation." 2023 IEEE International Conference on Robotics and Automation (ICRA). IEEE, 2023.
> >
> > [8] Janner, Michael, et al. "Planning with diffusion for flexible behavior synthesis." arXiv preprint arXiv:2205.09991 (2022).
> >
> > [9] Yang, Brian, et al. "Diffusion-es: Gradient-free planning with diffusion for autonomous driving and zero-shot instruction following." *arXiv preprint arXiv:2402.06559* (2024).
> >
> > [10] Sun, Qiao, et al. "Large Trajectory Models are Scalable Motion Predictors and Planners." *arXiv preprint arXiv:2310.19620* (2023).

---

> > > ### Comment · Reviewer_hoYE · 2024-11-23
> > >
> > > Thanks for the detailed response of my questions by the authors. Diffusion seems to be a promising realm in tackling the motion planning problem for AV stack. Hence, I would raise my ratings accordingly. It is also encouraged to provide extra results in compuational cost/ inference time, etc. for better understanding. Also, it would be better if the authors consider open-source this project afterwards, which may provide great help in the community.

---

> > > > ### Author Response · Authors · 2024-11-23
> > > > **Thanks for increasing to 8!**
> > > >
> > > > Thank you so much for your effort engaged in the review phase and for increasing to 8! We have included the inference time and comparisons with other diffusion-based algorithms in Appendix B. Additionally, we are actively working on open-sourcing our code and data. Thanks again！

---

### Official Review · Reviewer_Csii · 2024-11-03

**Soundness:** 4
**Presentation:** 3
**Contribution:** 4
**Rating:** 8
**Confidence:** 4

**Summary:**

This paper presents a novel approach for autonomous driving using a transformer-based Diffusion Planner. By leveraging diffusion models, the authors address key challenges in autonomous driving, such as modeling multi-modal driving behavior and ensuring trajectory quality without relying on rule-based refinement. The approach demonstrates state-of-the-art performance on the nuPlan benchmark and a newly collected dataset.

**Strengths:**

The strengths of the paper are threefold.

1. The use of diffusion models in the autonomous driving planning task is novel. The authors effectively address the limitations of existing learning-based planning methods, such as handling multi-modal behaviors and out-of-distribution scenarios.

2. The paper provides a thorough explanation of the Diffusion Planner’s architecture and how it integrates prediction and planning tasks. The classifier guidance mechanism for adaptable planning behaviors is particularly well-explained.

3. The evaluations on the nuPlan benchmark and the delivery-vehicle dataset demonstrate impressive closed-loop performance, surpassing both learning-based and rule-based baselines. The method shows robust transferability, highlighting its potential for real-world applications.

**Weaknesses:**

The overall quality of this paper is strong. One minor area for improvement is the quantity of simulation benchmarks used. The paper evaluates planner performance solely based on the Test14 random benchmark, which consists of approximately 280 closed-loop scenarios. It would be beneficial to include additional simulation benchmarks from nuPlan, such as the Val14 benchmark and the Test14 hard benchmark, to provide a more comprehensive demonstration of the advantages of the proposed method over the baselines.

**Questions:**

The suggestion of improvement is stated in the *Weaknesses*. No further suggestions.

---

> ### Author Response · Authors · 2024-11-19
> **Response to Reviewer Csii**
>
> We really appreciate the reviewer for the constructive comments and positive feedback on our paper.
>
> > **W1. It would be beneficial to include additional simulation benchmarks from nuPlan, such as the Val14 benchmark and the Test14 hard benchmark**
>
> We have added experimental results on the Val14 and Test14-Hard benchmarks, where **we continued to achieve state-of-the-art performance among learning-based methods**. By incorporating a refinement step, we were able to achieve a score above 94 (Val14 and Test14), surpassing human expert level. Please refer to the new results in the General Response and Appendix B of our revised paper.

---

### Official Review · Reviewer_J7se · 2024-11-04

**Soundness:** 3
**Presentation:** 3
**Contribution:** 2
**Rating:** 6
**Confidence:** 4

**Summary:**

Current learning-based planning methods, such as imitation learning, often struggle with safety and adaptability, especially when dealing with the multi-modal behaviors typical of human drivers. To overcome these limitations, the authors introduce a novel transformer-based Diffusion Planner designed for closed-loop planning. This model captures multi-modal driving behaviors and ensures high-quality trajectories without relying on rule-based refinements. They integrate prediction and planning tasks within the same architecture to facilitate cooperative vehicle behaviors. Additionally, the Diffusion Planner enhances safety and adaptability by learning trajectory score gradients and utilizing a flexible classifier guidance mechanism.

**Strengths:**

S1. Reduces complexity  issue by collectively considering the status of key participants in the driving scenario and jointly modeling the motion prediction and closed-loop planning tasks as a future trajectory generation task.

S2. Integrating closed-loop planning with a diffusion model is an effective approach, and the use of the architecture is clearly articulated.

**Weaknesses:**

W1. Though PLUTO (hybrid method) performs better than Diffusion Planner in the NuPlan dataset, no comparison with PLUTO w or w/o refine is shown for the delivery-vehicle driving dataset.

W2. The paper would benefit from a more explicit and detailed statement of contributions, perhaps in a dedicated paragraph near the end of the introduction. This should clearly outline how the Diffusion Planner addresses each of the limitations mentioned and what specific novel aspects it introduces.

Minor Nitpicks

N1. A legend should be added to Appendix A, Figure 8
N2. Line 195: Conditions C could be mathematically defined

**Questions:**

Same as weaknesses. Additional questions:

Q1. Line 466-477, “Due to space limitations, …”, what space limitations is the author mentioning?

Q2. Could the authors elaborate on W1 and explain why PLUTO (with and without refinement) was not included in the comparisons on the delivery-vehicle driving dataset, given its strong performance on NuPlan?

---

> ### Author Response · Authors · 2024-11-19
> **Response to Reviewer J7se**
>
> We thank the reviewer for the constructive comments. Regarding the concerns of the reviewer J7se, we provide the following responses.
>
> > **W1. & Q3. Though PLUTO (hybrid method) performs better than Diffusion Planner in the NuPlan dataset, no comparison with PLUTO w or w/o refine is shown for the delivery-vehicle driving dataset**
>
> - We thank the reviewer for the valuable feedback. The PLUTO pipeline involves a specific and time-consuming data processing stage. We have made our best efforts to go through the entire PLUTO pipeline using the delivery vehicle datasets; however, it requires significant CPU resources and the provided parallel processing utility often encounters errors, causing interruptions in data conversion. Given the current progress, it is unlikely that we will be able to complete all training by the end of the rebuttal period. However, we will add results in the final version.
> - We are able to report PLUTO’s results on the nuPlan dataset thanks to the official checkpoint. **Additionally, we have included results from the larger Val14 and more challenging Test14-hard benchmarks to highlight the advantages of our approach over PLUTO. Please refer to the General Response or Appendix B. for more details**.
>
> > **W2. This should clearly outline how the Diffusion Planner addresses each of the limitations mentioned and what specific novel aspects it introduces**
>
> We appreciate the reviewer's constructive comments. We have added a detailed statement of contributions at the end of the introduction, which we briefly outline here:
>
> - To the best of authors' knowledge, this paper is the first to fully harness the power of diffusion models with a specifically designed architecture for high-performance motion planning, without reliant on rule-based refinement, as demonstrated in the paper and General Response GR1.
> - As shown in General Response GR1,2, our experiments in the paper, and Appendix B, we achieve SOTA closed-loop performance on nuPlan benchmark, generating more robust and smoother trajectories compared to the baselines, even without post-refinement, as illustrated in Figure 3.
> - We demonstrate that our model can achieve personalized and flexible driving behavior at runtime by utilizing a guidance mechanism—a desirable feature for real-world applications that is technically challenging for existing transformer-based planning methods.
> - We have collected and evaluated a new 200-hour delivery-vehicle dataset, carefully processing it to ensure compatibility with the nuPlan framework. We will open-source this dataset as a supplement to nuPlan.
>
> > **N1. A legend should be added to Appendix A, Figure 8 N2. Line 195: Conditions C could be mathematically defined**
>
> - We appreciate the reviewer’s reminder. The meanings of the elements in Figure 8 have been introduced in Figures 1 and 3 of the main text. In the latest version, we have added the same explanations for Figure 8.
> - For the mathematical definition of condition C, we have actually provided the explanation in Section 4.2, including its definition, dimensions, and the formula for how it is integrated into the network.
>
> > **Q1. Line 466-477, “Due to space limitations, …”, what space limitations is the author mentioning**
>
> Sorry for the confusion. We actually meant "space limit." We intended to highlight that various forms of guidance and combinations can be explored in practical applications. Due to the space constraints of our paper, we selected three types of guidance to showcase in the experimental section. We have revised the paper to make this clearer.
>
>
> > **Q2. It is unclear in Figure 7 about the choice of hyperparameters in the proposed method. Could the author elaborate on the seen trends in Figure 7.**
>
> We apologize for the unclear illustration. For a more detailed analysis of Figure 7, please refer to Section 5.2 (lines 512-517). In Figure 7, darker colors indicate better performance. We find that a low temperature is necessary, and the model demonstrates good robustness across various step counts and levels of low temperature. Ultimately, we select 25 steps for denoising and set the low-temperature sampling parameter to 0.5, as shown in Table 6 in Appendix C.2.

---

> > ### Author Response · Authors · 2024-11-24
> > **Sorry to bother you**
> >
> > Dear reviewer J7se,
> >
> > As the discussion period is coming to a close, we wanted to check back to see whether you have any remaining questions. **We would be happy to clarify further, and grateful for any other feedback you may povide**. We really appreciate your time engaged in the review and rebuttal phase.
> >
> > Thank you very much and look forward to your replies!
> >
> > Best regards,
> >
> > Authors of Paper 13578

---

> > ### Comment · Reviewer_J7se · 2024-12-03
> >
> > Thank you for your response. I appreciate you considering the suggestions and incorporating them into the paper.
> >
> > It's disappointing to hear that the implementation of PLUTO could not be completed within the expected time frame. I remain unconvinced about the solution's effectiveness due to the absence of benchmarks on real-world data and its failure to outperform the SOTA on the NuPlan dataset. Therefore, I will maintain my rating.

---

> > > ### Author Response · Authors · 2024-12-03
> > > **Response to Reviewer J7se**
> > >
> > > Thank you for the reviewer’s response.
> > >
> > > Regarding the Pluto results on the delivery vehicle dataset, we have made our best effort during the limited rebuttal period. **We used two 1.5TB, 384-CPU servers, but only completed 1/3 data conversions during the entire rebuttal period (~20days)**. We used the official code, but due to the heavy CPU load, frequent interruptions occurred. **On average, we had to restart the code every three hours**.
> > >
> > > We respectfully disagree with the reviewer’s statement that there has been a "failure to outperform the SOTA on the NuPlan dataset." To demonstrate this, we have included evaluations on the larger Val14 and more challenging Test14-hard benchmarks. The results show that Diffusion Planner has achieved SOTA performance across both reactive and non-reactive evaluation settings. Please refer to the general response and Appendix B for further details. We report them again as follows:
> > >
> > > | Methods                  | Val14 (NR) | Val14 \(R\) | Test14 (NR) | Test14 \(R\) | Test14-hard (NR) | Test14-hard \(R\) |
> > > | ------------------------ | ---------- | ----------- | ----------- | ------------ | ---------------- | ----------------- |
> > > | PLUTO w/o refine.*       | 88.89      | 78.11       | **89.90**   | 78.62        | 70.03            | 59.74             |
> > > | Diffusion Planner (ours) | **89.76**  | **82.56**   | **89.22**   | **83.36**    | **75.67**        | **68.56**         |
> > >
> > > | Methods                              | Val14 (NR) | Val14 \(R\) | Test14 (NR) | Test14 \(R\) | Test14-hard (NR) | Test14-hard \(R\) |
> > > | ------------------------------------ | ---------- | ----------- | ----------- | ------------ | ---------------- | ----------------- |
> > > | **Expert (Log-replay)**              | 93.53      | 80.32       | 94.03       | 75.86        | **85.96**        | 68.80             |
> > > | PLUTO                                | 92.88      | 76.88       | 92.23       | 90.29        | 80.08            | 76.88             |
> > > | Diffusion Planner with refine (ours) | **94.26**  | **92.90**   | **94.80**   | **91.75**    | 78.87            | **82.00**         |

---

> > > > ### Comment · Reviewer_J7se · 2024-12-03
> > > >
> > > > I still have some reservations regarding the results, as the performance improves on larger datasets but tends to be less favorable on smaller ones.
> > > >
> > > > Thank you for your efforts during the rebuttal period and for attempting to run the PLUTO algorithm. I understand the limitations that prevented you from running it, but it would have been helpful to see those results.
> > > >
> > > > After thoughtful reconsideration and reviewing the responses to other reviewers, I have decided to raise the rating to a 6!

---

> > > > ### Author Response · Authors · 2024-12-03
> > > > **Response to Reviewer J7se**
> > > >
> > > > Thank you so much for your effort in the review phase and for increasing the score to 6! Your valuable feedback greatly improved our paper! Regarding the missing evaluation of Pluto on the delivery dataset, we will continue working on those experiments and report the results in our final version.
> > > >
> > > > Additionally, we would like to kindly point out that the version of Pluto without refinement (Pluto w/o refine) used strong priors as model inputs (pre-searching the centerline), but only achieved a 0.7-point higher score than us on test14. We can surpass Pluto w/o refine by a large margin (~+2) by introducing a guidance mechanism. Specifically, by simultaneously applying collision and drivable guidance, we can achieve a score of 92, which is even comparable to Pluto with refinement. Furthermore, our version with refinement outperforms Pluto in most cases.
> > > >
> > > > Thanks again for your constructive comments!

---

> ### Comment · Area_Chair_mK5A · 2024-11-27
>
> Dear Reviewer,
>
> Could you please look at the author feedback?
>
> Thanks,
> AC

---

### Author Response · Authors · 2024-11-19
**General Response 1/2**

We thank all the reviewers for the effort engaged in the review phase and the constructive comments.

**We have revised our paper (highlighted in blue text color). The modifications are summarized as follows.**
1. (For Reviewer J7se, Csii, hoYE, d65s). We add more extensive results on Val14 and Test14-hard benchmarks and compare them with existing diffusion-based planners in Appendix B.
2. (For Reviewer J7se, d65s). We add a detailed statement of contributions at the end of the introduction, Section 1.
3. (For Reviewer hoYE, d65s). We move the discussion of diffusion-based methods and additional traffic simulation-related work to the "Related Works", Section 2.
4. (For Reviewer J7se). We add additional explanations in Sections 5.1, 5.2, and Appendix A in response to the reviewer's request for further clarification.
5. (For Reviewer d65s). We add a case study on the use of combined collision and drivable guidance in Appendix F, along with a discussion on energy function design in Appendix C.3.

**Regarding the reviewers' common comments, we provide the following general responses**.
> **GR1. Performance on Val14 and Test14-hard in both non-reactive and reactive modes (without refine).**

We conduct additional experiments on Val14 and Test14-hard closed-loop non-reactive (NR) and reactive \(R\) results for all learning-based baselines as follows. **We have also included two recent works that use diffusion for planning [1] [2] ([1] borrows existing diffusion model with LLM, [2] released after our submission), with refinements or LLMs removed, to demonstrate the advantages of our model over existing diffusion-based methods**.

| Methods                         | Val14 (NR) | Val14 \(R\) | Test14-hard (NR) | Test14-hard \(R\) |
| ------------------------------- | ---------- | --------- | ---------------- | --------------- |
| PDM-Open*                       | 53.53      | 54.24     | 33.51            | 35.83           |
| UrbanDriver                     | 68.57      | 64.11     | 50.40            | 49.95           |
| GameFormer w/o refine.          | 13.32      | 8.69      | 7.08             | 6.69            |
| PlanTF                          | 84.72      | 76.95     | 69.70            | 61.61           |
| PLUTO w/o refine.*              | 88.89      | 78.11     | 70.03            | 59.74           |
| [1] Diffusion-es w/o LLM        | 50.00      | -         | -                | -               |
| [2] STR2-CPKS-800M w/o refine.* | 8.80       | -         | 10.99            | -               |
| Diffusion Planner (ours)        | **89.76**  | **82.56** | **75.67**        | **68.56**       |
*: Using pre-searched reference lines or additional proposals as model inputs provides prior knowledge.

---

> ### Author Response · Authors · 2024-11-19
> **General Response 2/2**
>
> > **GR2. Experiments with extra post-processing**
>
> To evaluate our methods with similar rule-based refinement, we directly applied an existing refinement module [2] after our learning-based method without any parameter tuning, which adjusts the model outputs. All trajectories were scored as described in [3]. We report the Val14, Test14, and Test14-Hard closed-loop non-reactive (NR) and reactive \(R\) results for all rule-based and hybrid methods, as follows.
>
>
>
> | Methods                             | Val14 (NR) | Val14 \(R\) | Test14-hard (NR) | Test14-hard \(R\) | Test14 (NR) | Test14 \(R\) |
> | ----------------------------------- | ---------- | --------- | ---------------- | --------------- | ----------- | ---------- |
> | **Expert (Log-replay)**             | 93.53      | 80.32     | **85.96**            | 68.80           | 94.03       | 75.86      |
> | IDM                                 | 75.60      | 77.33     | 56.15            | 62.26           | 70.39       | 74.42      |
> | PDM-Closed                          | 92.84      | 92.12     | 65.08            | 75.19           | 90.05       | 91.63      |
> | PDM-Hybrid                          | 92.77      | 92.11     | 65.99            | 76.07           | 90.10       | 91.28      |
> | GameFormer                          | 79.94      | 79.78     | 68.70            | 67.05           | 83.88       | 82.05      |
> | PLUTO                               | 92.88      | 76.88     | 80.08        | 76.88           | 92.23       | 90.29      |
> | [1] Diffusion-es                    | 92.00      | -         | -                | -               | -           | -          |
> | [2] STR2-CPKS-800M                  | 93.91      | 92.51     | 77.54            | **82.02**       | -           | -          |
> | Diffusion Planner with refine (ours) | **94.26**  | **92.90** | 78.87            | **82.00**       | **94.80**   | **91.75**  |
>
> Our method, with the addition of post-processing, achieved scores that other methods could not reach, even surpassing human performance. **This is due to our model's ability to output high-quality trajectories, further enhanced by post-processing**. We have visualized the trajectory quality in Figure 3. It should also be noted that, as shown in the previous table in GR1, many high-scoring methods experience significant score drops when refinement is removed. **This highlights their over-reliance on refinement**, which is consistent with our discussions in the introduction and related work sections.
>
>
> [1] Yang, Brian, et al. "Diffusion-es: Gradient-free planning with diffusion for autonomous driving and zero-shot instruction following." *arXiv preprint arXiv:2402.06559* (2024).
>
> [2] Sun, Qiao, et al. "Generalizing Motion Planners with Mixture of Experts for Autonomous Driving." *arXiv preprint arXiv:2410.15774* (2024).
>
> [3] Dauner, Daniel, et al. "Parting with misconceptions about learning-based vehicle motion planning." *Conference on Robot Learning*. PMLR, 2023.

---

### Meta-Review · Area_Chair_mK5A · 2024-12-23

**Metareview:**

This work proposes a novel transformer-based Diffusion Planner for closed-loop planning, which could resolve current bottlenecks in prior works. The structure is neat, without rule-based refinement and could effectively model the multi-modal driving behavior. Evaluations on large-scale benchmarks, both the nuPlan and the self-collected 200-hour (which is impressive in terms of public data), prove the efficacy of the proposed method.

This paper provides solid experiments with good motivation. The paper is well written. The proposed method is detailed with clear implementation on nuPlan for the closed-loop verification. Diffusion is a good methodology to be adopted for the autonomous driving community, with many potential applications for video generation, decision making, motion prediction. Many applications in the autonomous driving domain. Such a work would shed new light on the community, and deserve a broad attention to be presented at ICLR.

Please incorporate all the necessary comments raised by reviewers and supplement new (unfinished during rebuttal, if any) experiments to make the submission more convincing for the camera-ready.

**Additional Comments On Reviewer Discussion:**

Authors have did an excellent job to address the reviewer comments. The final rating were (8 8 8 6). Apart from the many good merits of the paper, the major concerns were:

- PLUTO method performance on the self-collected daetaset
- Re-write / re-organize some part in the introduction to make the contribution (i.e. diffusion) more explicit and straightforward.
- More data / test scenarios to verify the proposed idea
- How to differentiate from existing works / more extensive review. Note: position the work without **overclaiming** the novelty.

Although the requested results of Pluto are not provided, authors are honest to provide experimental details as to why they fail to do so. Please incorporate the full experiments in the future and release the dataset for the sake of the community.

AC has carefully checked the back-and-forth discussions between reviewers and authors. Most concerns were addressed well. Three out of four reviewers are actively engaged in the review process.

---

### Decision · Program_Chairs · 2025-01-22

Accept (Oral)